# Zinc homeostasis governed by Golgi-resident ZnT family members regulates ERp44-mediated proteostasis at the ER-Golgi interface

Yuta Amagai [1], Momo Yamada[2], Toshiyuki Kowada [1,2,3], Tomomi Watanabe[3], Yuyin Du[4], Rong Liu[3], Satoshi Naramoto[5,9], Satoshi Watanabe [1,2,3], Junko Kyozuka [5], Tiziana Anelli[6], Tiziana Tempio[6], Roberto Sitia [6], Shin Mizukami[1,2,3,7] & Kenji Inaba [1,2,3,7,8] ✉

Many secretory enzymes acquire essential zinc ions ($Zn^{2+}$) in the Golgi complex. ERp44, a chaperone operating in the early secretory pathway, also binds $Zn^{2+}$ to regulate its client binding and release for the control of protein traffic and homeostasis. Notably, three membrane transporter complexes, ZnT4, ZnT5/ZnT6 and ZnT7, import $Zn^{2+}$ into the Golgi lumen in exchange with protons. To identify their specific roles, we here perform quantitative $Zn^{2+}$ imaging using super-resolution microscopy and $Zn^{2+}$-probes targeted in specific Golgi subregions. Systematic ZnT-knockdowns reveal that ZnT4, ZnT5/ZnT6 and ZnT7 regulate labile $Zn^{2+}$ concentration at the distal, medial, and proximal Golgi, respectively, consistent with their localization. Time-course imaging of cells undergoing synchronized secretory protein traffic and functional assays demonstrates that ZnT-mediated $Zn^{2+}$ fluxes tune the localization, trafficking, and client-retrieval activity of ERp44. Altogether, this study provides deep mechanistic insights into how ZnTs control $Zn^{2+}$ homeostasis and ERp44-mediated proteostasis along the early secretory pathway.

Zinc is an essential catalytic or structural cofactor for ~10% of the human proteome[1]. Since zinc ions ($Zn^{2+}$) mediate key processes in virtually all subcellular compartments, disturbances in their homeostasis cause a wide spectrum of diseases[2,3]. In eukaryotic cells, two transporter families play pivotal roles for the maintenance of $Zn^{2+}$ homeostasis. Members of the ZnT (zinc transporter, SLC30) family mobilize $Zn^{2+}$ from the cytosol into the extracellular space or the lumen of intracellular organelles, in exchange with protons. Meanwhile, ZIP (Zrt-/Irt-like protein, SLC39) family members transport $Zn^{2+}$ in opposite directions. Their redundancy and different subcellular distribution suggest a tight regulation of $Zn^{2+}$ fluxes in cells and organelles[2,4].

[1]Institute of Multidisciplinary Research for Advanced Materials, Tohoku University, Sendai, Miyagi 980-8577, Japan. [2]Department of Chemistry, Graduate School of Science, Tohoku University, Sendai, Miyagi 980-8578, Japan. [3]Department of Molecular and Chemical Life Sciences, Graduate School of Life Sciences, Tohoku University, Sendai, Miyagi 980-8577, Japan. [4]Department of Chemistry, Faculty of Science, Tohoku University, 6-3 Aramaki-aza-Aoba, Aoba-ku, Sendai, Miyagi 980-8578, Japan. [5]Department of Ecological Developmental Adaptability Life Sciences, Graduate School of Life Sciences, Tohoku University, Sendai, Miyagi 980-8577, Japan. [6]Division of Genetics and Cell Biology, Vita-Salute University, IRCCS Ospedale San Raffaele, 20132 Milan, Italy. [7]Core Research for Evolutional Science and Technology (CREST), Japan Agency for Medical Research and Development (AMED), Chiyoda, Tokyo, Japan. [8]Medical Institute of Bioregulation, Kyushu University, Fukuoka 812–8582, Japan. [9]Present address: Department of Biological Sciences, Faculty of Science, Hokkaido University, Kita 10 Nishi 8, Kita-ku, Sapporo, Japan. ✉e-mail: kenji.inaba.a1@tohoku.ac.jp

The early secretory pathway (ESP) comprising the endoplasmic-reticulum (ER) and the Golgi apparatus is the space where newly synthesized secretory proteins acquire their native structure and undergo key post-translational modifications, including metalation. The ESP harbors three dimeric ZnT (ZnT4, ZnT5/ZnT6, and ZnT7) and four ZIP members (ZIP7, ZIP9, ZIP11, and ZIP13) that may regulate the labile $Zn^{2+}$ concentration (hereafter referred to as $[Zn^{2+}]$) and fluxes[2]. Their pharmacologic or genetic inhibition causes severe cellular and organismal problems[2,5–7], emphasizing the physiological importance of $Zn^{2+}$ in the ESP. Yet, the function and regulation of these ZnT and ZIP family members are still poorly understood.

Endoplasmic reticulum protein 44 (ERp44) is a member of the protein disulfide isomerase (PDI) family, consisting of three thioredoxin-like domains and a C-terminal tail (C-tail) that ends with an RDEL ER-localization motif. ERp44 cycles between the ER and Golgi, controlling the localization of ER-resident enzymes that lack KDEL motifs (e.g. ER oxidoreductin 1α (Ero1α), ER aminopeptidase 1 (ERAP1), Peroxiredoxin 4 (Prx4)) and the folding of complex secretory cargo proteins (IgM, adiponectin or IL-12 subunits)[8]. ERp44 captures its clients in post-ER compartments, mostly forming disulfide-linked complexes via Cys29 in its active site, and retrieves them to the ER via KDEL receptors (KDELR)[9–16]. Of note, our recent studies revealed that $Zn^{2+}$ has a crucial role in regulating the structure and function of ERp44. Binding of $Zn^{2+}$ to its histidine cluster located at the base of the C-tail triggers massive domain rearrangements that simultaneously exposes the substrate-binding surface for efficient client capture[17]. Although ZnT5, ZnT6, and ZnT7 are predicted to supply $Zn^{2+}$ to ERp44 in the Golgi[17], their precise roles in the regulation of ERp44-dependent protein quality control, and more in general in the maintenance of $Zn^{2+}$ homeostasis in the ESP, remain to be elucidated.

For the purpose of quantitative $Zn^{2+}$ imaging, we recently developed an organelle-targetable $Zn^{2+}$-fluorescent probes, ZnDA-1H, ZnDA-2H and ZnDA-3H. These probes allowed us to determine that the $[Zn^{2+}]$ is much greater in the Golgi ($25 \pm 1$ nM) than in the ER (14 pM)[18,19]. Here we exploit this methodology to determine in detail the $[Zn^{2+}]$ in the different Golgi subregions (i.e. cisternae). Combining $[Zn^{2+}]$ measurements with systematic ZnT knockdowns, moreover, we characterize the roles of each ZnT complex in $Zn^{2+}$ uptake into the different Golgi cisternae. In this connection, we observe that their individual or combined knockdown differently affects the subcellular localization, client retrieval activity, and ER-to-Golgi traffic of ERp44. Altogether, our results provide detailed mechanistic insights on how the different ZnTs control $Zn^{2+}$ homeostasis along the ESP and ERp44-mediated proteostasis at the ER-Golgi interface.

## Results

### Targeting of $Zn^{2+}$ probe to different subregions of the Golgi apparatus

We recently developed an organelle-targetable $Zn^{2+}$ probe named ZnDA-1H, establishing the methodology for quantifying the $[Zn^{2+}]$ in the secretory pathway. The probe can be targeted, via its HaloTag ligand (HTL) moiety, to a desired organelle where a Halo-tagged protein is expressed as a fusion with a protein of known intracellular localization. Analysis of HeLa cells expressing a HaloTag-mannosidase-II (ManII) fusion protein revealed that the $[Zn^{2+}]$ in the Golgi ($[Zn^{2+}]_{Golgi}$) was $25 \pm 1$ nM[18,20]. To verify whether $[Zn^{2+}]$ varies in the different Golgi subregions, and to explore the role of the three main Golgi-resident ZnTs, we constructed Halo-tagged ERGIC-53 (ER Golgi intermediate compartment protein 53), ManII, TPST2 (tyrosyl protein sulfotransferase 2), and GalT (β−1,4-galactosyltransferase 1) proteins and expressed them in HeLa Kyoto cells. Considering the predicted topology models of the above marker proteins, HaloTag was inserted immediately after the signal peptide of ERGIC-53, and into the C-termini of ManII, TPST2 and GalT so that HaloTag is located in the luminal side of the Golgi apparatus. Immunofluorescence analyses

revealed that, as expected, all Halo-tagged proteins largely localized in the Golgi area, overlapping with GM130, a *cis*-Golgi marker (Supplementary Fig. 1). To determine their localization with higher accuracy, we labeled the expressed Halo-tagged proteins with HaloTag-TMR ligand (HTL-TMR), and treated cells with nocodazole to create Golgi mini-stacks. The latter are believed to conserve the native-like multicisternal structure and transport functions of the Golgi[21–23] and allow higher resolution imaging. Subsequently, cells were fixed, immunostained for GM130 (*cis*-Golgi marker), giantin (medial-Golgi marker), ManII (*cis*-/medial-Golgi marker), and Golgin97 (*trans*-Golgi network (TGN) marker[22]), and observed by Airyscan super-resolution microscopy. By using chromatic shift-corrected fluorescence images[24], we compared the signal positions of the Halo-tagged proteins relative to the markers (Fig. 1a, b). In this way, we verified the specific locations of Halo-ERGIC-53 in the immediate upstream of the *cis*-Golgi, hereafter called pre-*cis*-Golgi, ManII-Halo in the *cis*-/medial-Golgi, Halo-TPST2 in the medial-Golgi, and GalT-Halo in the *trans*-Golgi (Fig. 1a–c).

To determine the $[Zn^{2+}]$ in each Golgi cisterna with a different pH-value, we determined the $K_d$ for $Zn^{2+}$ of the HaloTag-ZnDA-1H conjugate at various pH values, taking into consideration the possible pH-dependent changes of $Zn^{2+}$-binding affinity of the probe[18]. To this end, fluorescence intensity excited at 440 nm was measured at different pH and $Zn^{2+}$ concentrations in vitro. We thereby determined that $K_d$ for $Zn^{2+}$ of the conjugate varied from $0.30 \pm 0.02\,\mu M$ at pH 7.4 to $1.6 \pm 0.1\,\mu M$ at pH 5.5 (Supplementary Fig. 2A, B).

For analyzing the pH in each Golgi cisterna of living HeLa Kyoto cells, pHluorin2, a ratiometric pH-sensitive fluorescent protein-based probe[25,26] was employed. We constructed and expressed pHluorin2-tagged STIM1(NN), ERGIC-53, ManII, TPST2, and GalT to measure pH in the ER, pre-*cis*-, *cis*-/medial-, medial-, and *trans*-Golgi, respectively. Cells were excited at 405 nm and 473 nm, and the fluorescent signal intensities, $F_{405}$ and $F_{473}$, were measured with laser scanning confocal microscopy. To gain a calibration plot of fluorescence signal intensity against pH, bacterially expressed MBP-pHluorin2 was immobilized on agarose beads, incubated at different pH, and then its fluorescence parameters were acquired under conditions identical to those used in live-cell imaging experiments (Supplementary Fig. 2C, D). By contrasting the $F_{405}$-to-$F_{473}$ ratios of pHluorin2 in cells to those on beads, we determined the pH values in the ER, pre-*cis*-, *cis*-/medial-, medial-, and *trans*-Golgi of HeLa Kyoto cells to be $7.11 \pm 0.08$, $7.02 \pm 0.11$, $6.53 \pm 0.13$, $6.51 \pm 0.07$, and $6.13 \pm 0.12$, respectively (Supplementary Fig. 2E, F).

### Golgi-resident ZnT family members regulate $Zn^{2+}$ concentrations in the different Golgi cisternae

Using the pH values thus obtained, we quantified the $[Zn^{2+}]$ in different Golgi cisternae under normal and ZnT4-, ZnT5/6-, and/or ZnT7-knockdown conditions. Quantitative real-time PCR (qRT-PCR) verified that each siRNA markedly and specifically reduced the mRNA levels of targeted genes (Supplementary Fig. 3A). Although the ZnT4-mRNA knockdown efficiency was moderate, the levels of exogenously expressed ZnT4-FLAG proteins were dramatically diminished in siZnT4 cells (Supplementary Fig. 3B). Given that ZnTs function as $Zn^{2+}/H^+$ antiporters[27–29], we further examined a possibility of Golgi acidification by ZnTs silencing. Our analyses with the pHluorin2-tagged proteins, however, indicated that the pH values were not significantly altered at any Golgi cisterna in the ZnT4-, ZnT5/6-, and/or ZnT7-knockdown cells compared to the normal (siControl) cells (Supplementary Fig. 2G, H).

As shown in Fig. 1d, our quantitative analyses revealed that, under the normal (siControl) conditions, the $[Zn^{2+}]$ in the pre-*cis*- (labeled by Halo-ERGIC-53), *cis*-/medial- (by ManII-Halo), medial (by TPST2-Halo), and *trans*-Golgi (by GalT-Halo) cisternae were $100 \pm 7$ nM (Means $\pm$ SEM, $n = 23$), $82 \pm 5$ nM ($n = 35$), $57 \pm 5$ nM ($n = 36$), and $84 \pm 5$ nM ($n = 36$), respectively. Statistical analyses confirmed that the medial-Golgi contains lower $[Zn^{2+}]$ than the other Golgi cisternae

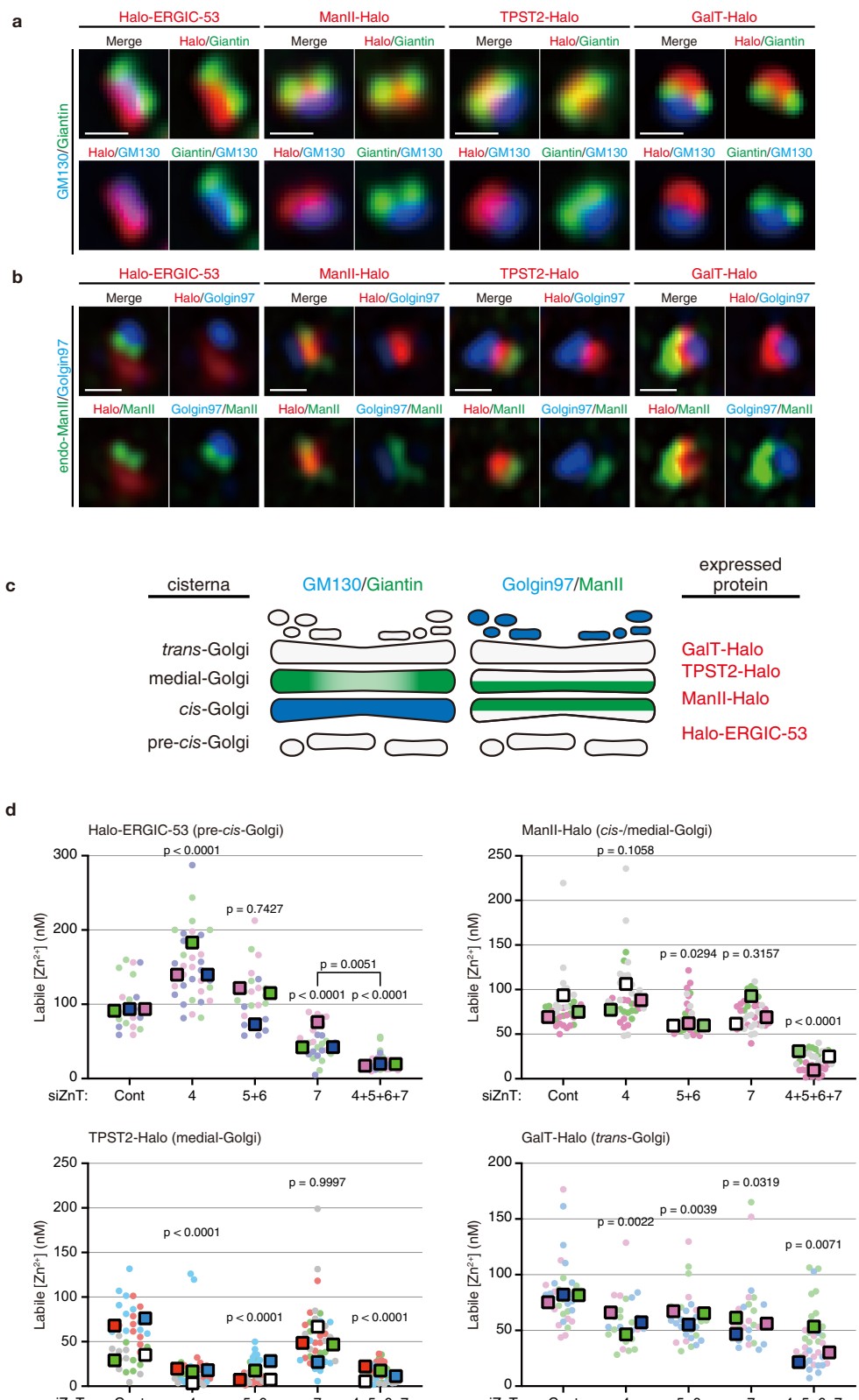

(Supplementary Fig. 4). In contrast, the [Zn²⁺] in the pre-*cis*-Golgi ($[\text{Zn}^{2+}]_{\text{pre-}cis}$) tends to be higher than in the other Golgi cisternae although the differences between the pre-*cis*- and *cis*-Golgi and between the pre-*cis*- and *trans*-Golgi are not statistically substantial (Supplementary Fig. 4).

We next investigated the consequences of knocking down the three ZnTs, individually or in combinations. As expected, the

simultaneous knockdown of ZnT4, ZnT5/6, and ZnT7 (i.e. siZnT4/5/6/7) markedly decreased the [Zn²⁺] in all Golgi cisternae (Fig. 1d). Similarly, the triple knockdown of ZnT5/6/7 decreased the [Zn²⁺] in all Golgi cisternae (Supplementary Fig. 5), suggesting that ZnT5/6/7 predominantly govern zinc homeostasis in the Golgi apparatus. Of note, the single knockdown of ZnT4, ZnT5/6 and ZnT7 significantly reduced the [Zn²⁺] in the *trans*-/medial-, *cis*-/medial-/*trans*-, and pre-*cis*-Golgi,

**Fig. 1 | ZnT-mediated control of $Zn^{2+}$ concentrations in the Golgi subcompartments. a** Localization of Halo-tagged marker proteins in Golgi mini-stacks. HeLa Kyoto cells transfected with indicated Halo-proteins were incubated with HTL-TMR (red), treated with nocodazole, and fixed. Fixed cells were co-stained for *cis*-Golgi (GM130; blue) and medial-Golgi (giantin; green). Fluorescence images were obtained by Airyscan super-resolution microscopy. Scale bars, 500 nm. Similar results were obtained from two independent experiments. **b** HeLa Kyoto cells were prepared as in (**a**) and co-stained for *cis*-/medial-Golgi (mannosidase II, ManII; green) and TGN (Golgin97; blue). Fluorescence images were obtained using Airyscan2 super-resolution microscopy. Scale bars, 500 nm. Similar results were obtained from two independent experiments. **c** Schematic representation of the localization of Halo-tagged Golgi marker proteins. **d** Labile $Zn^{2+}$ concentration ($[Zn^{2+}]$) in the Golgi cisternae measured by ZnDA-1H. HeLa Kyoto cells were transfected with the indicated siRNAs and Halo-tagged Golgi marker proteins. $[Zn^{2+}]$ under each knockdown condition was determined as described in *Methods*. A circular dot indicates each datapoint, and a rectangle dot indicates the median of each experiment. Data for Halo-ERGIC-53 (siCont, $n = 23$ cells; siZnT4, $n = 35$ cells; siZnT5 + 6, $n = 25$ cells; siZnT7, $n = 30$ cells; siZnT4 + 5 + 6 + 7, $n = 33$ cells), ManII-Halo (siCont, $n = 24$ cells; siZnT4, $n = 28$ cells; siZnT5 + 6, $n = 27$ cells; siZnT7, $n = 27$ cells; siZnT4 + 5 + 6 + 7, $n = 27$ cells), and GalT-Halo (siCont, $n = 36$ cells; siZnT4, $n = 25$ cells; siZnT5 + 6, $n = 33$ cells; siZnT7, $n = 26$ cells; siZnT4 + 5 + 6 + 7, $n = 38$ cells) were obtained from 3 independent experiments. Data for TPST2-Halo (siCont, $n = 36$ cells; siZnT4, $n = 33$ cells; siZnT5 + 6, $n = 50$ cells; siZnT7, $n = 40$ cells; siZnT4 + 5 + 6 + 7, $n = 40$ cells) were obtained from 4 independent experiments. One-way ANOVA followed by Tukey's multiple comparison test was used for statistical analysis. Source numerical data are provided in the Source Data file.

respectively (Fig. 1d). Unexpectedly, silencing ZnT4 increased the $[Zn^{2+}]_{pre\text{-}cis}$ while ZnT5/6-knockdown had only marginal effects (Fig. 1d). Thus, even though the mechanisms of the ZnT4-mediated zinc regulation remain elusive in part (see also the next paragraph), the above data collectively indicate that ZnT4 and ZnT5/6 are primarily involved in delivering $Zn^{2+}$ into the distal (i.e. medial-/*trans*-) Golgi, whereas ZnT7 is responsible for $Zn^{2+}$ uptake into more upstream regions (i.e. pre-*cis*-Golgi).

Considering the possibility that disrupting $Zn^{2+}$ homeostasis in the Golgi affects cytosolic $Zn^{2+}$ homeostasis, we also measured the cytosolic $[Zn^{2+}]$ ($[Zn^{2+}]_{cyto}$) by expressing a HaloTag protein with a nuclear export signal sequence (Halo-NES) and using ZnDA-2H, another zinc-detectable probe with higher affinity for $Zn^{2+}$ than ZnDA-1H[19]. The cytosolic $[Zn^{2+}]$ of HeLa Kyoto cells was estimated to be $0.46 \pm 0.14$ nM under normal conditions. Intriguingly, ZnT4-silencing increased it to $0.63 \pm 0.18$ nM, whereas ZnT5/6-, ZnT7-, and ZnT4/5/6/7-silencing reduced it to $0.17 \pm 0.09$ nM, $0.19 \pm 0.11$ nM, and $0.35 \pm 0.11$ nM, respectively (Supplementary Fig. 6A). To interpret these unexpected results, we investigated the effects of silencing these transporters on the mRNA levels of several zinc homeostasis genes, metallothionein 2 A (MT2A), ZnT1, and ZIP14. MT2A and ZnT1 are known to be upregulated when cells are treated with excess $Zn^{2+}$[30], whereas ZIP14 increases in cells upon zinc chelation with TPEN[31]. Our qRT-PCR analysis demonstrated that ZnT4-silencing upregulated MT2A transcription, whereas the other silencing conditions had marginal effects (Supplementary Fig. 6B). Thus, although the detailed mechanism is unknown, ZnT4 plays a significant role in regulating the $[Zn^{2+}]_{cyto}$, and the increased $[Zn^{2+}]_{cyto}$ under the ZnT4 knockdown condition likely leads to the increment of $[Zn^{2+}]_{pre\text{-}cis}$ (see Discussion).

### ZnT7 and ZnT6 localize at the different Golgi cisternae
Previous studies investigated the intracellular localization of ZnT6 and ZnT7[32,33]. Here we sought to quantitatively investigate the intra-Golgi distribution of ZnT5/6 and ZnT7 using super-resolution microscopy. The specificities of anti-ZnT6 and anti-ZnT7 antibodies were confirmed by immunoblotting and immunofluorescence (Supplementary Fig. 7A–D). ZnT6 and ZnT7 signals almost completely disappeared from the Golgi after the corresponding knockdowns (Supplementary Fig. 7B, D). Thus, endogenous ZnT6 and ZnT7 are primarily localized in the Golgi. ZnT6 forms heterodimers with ZnT5, and thereby localizes ZnT5 in the Golgi[34]. Thus, the ZnT6 signals likely correspond to ZnT5/6 heterodimers in the Golgi.

To determine the localization of ZnT6 and ZnT7 with higher accuracy, we created Golgi mini-stacks by nocodazole treatment, and analyzed them by super-resolution microscopy (Fig. 2a, c). Their cellular localization was classified into 5 groups (I–V or I'–V') based on their locations relative to GM130, Halo-ERGIC-53 or TPST2 (Fig. 2b, d). The results revealed that most ZnT6 (~80%) localized downstream of GM130 (class I and II in Fig. 2b). In contrast, more than 80% of ZnT7 was found upstream of GM130 and well overlapped with Halo-ERGIC-53 (class III, IV, and V in Fig. 2b). Additional analyses revealed that both ZnT6 and ZnT7 accumulated mostly upstream of TPST2-Halo (Class III', IV', and V' in Fig. 2d). Thus, ZnT6 mainly localizes in the *cis*- and medial-Golgi, whereas ZnT7 in the pre-*cis*- and *cis*-Golgi. The sub-cellular localizations of ZnT6 and ZnT7 were not altered by ZnT4-, ZnT5/6-, or ZnT7-silencing (Supplementary Fig. 8A–F), excluding the possibility of gross relocalizations of the ZnTs in response to $[Zn^{2+}]$ changes.

### Golgi-resident ZnTs regulate the intracellular localization of ERp44 differently
The different intracellular localization of ZnTs suggests that each family member has distinct physiological function. ERp44 reversibly redistributes between the ER and Golgi depending on $Zn^{2+}$ availability[17]. To explore which and how ZnT(s) regulate(s) ERp44 function, we first investigated the subcellular localization of ERp44 upon ZnT knockdown by calculating its Pearson's correlation coefficient with GM130 (Fig. 3). In control cells, ERp44 was mostly observed as dot-like structures widely in the cell and showed good colocalization with GM130 and endogenous ERGIC-53, indicating that ERp44 primarily localizes in the Golgi and ERGIC (Fig. 3a and Supplementary Fig. 9A). Additionally, ERp44 was also found in the ER and ERES (ER exit site) (Supplementary Fig. 9C), as reported previously[10]. In agreement with previous observations[17], the simultaneous knockdown of ZnT4, ZnT5, ZnT6 and ZnT7 relocated ERp44 to the Golgi area. Notably, the single knockdown of ZnT7 was sufficient to dislocate most ERp44 to the Golgi area (Fig. 3a, b). In contrast, the knockdown of ZnT4 or ZnT5/6 significantly reduced the localization of ERp44 in the Golgi and ERGIC (Fig. 3a, b, and Supplementary Fig. 9A, B). In these conditions, ERp44 appeared to be associated with exogenously expressed Halo-STIM1(NN), an ER marker, and endogenous SEC24C, an ERES marker (Supplementary Fig. 9C). Thus, ERp44 seems likely to be transported back to the ER and/or ERES more efficiently in ZnT4- and ZnT5/6-knockdown cells than in control cells. This observation is consistent with the increased $[Zn^{2+}]_{pre\text{-}cis}$ under the ZnT4-knockdown condition (Fig. 1d), while ZnT5/6-knockdown cells may exploit other mechanisms for promoting the retrograde transport of ERp44 to the ER.

In this connection, we investigated whether the altered $[Zn^{2+}]_{Golgi}$ affects the function of COPI and KDELR, essential components for the Golgi-to-ER retrograde transport. A slight increase of β-COP, a COPI coatomer protein, at the Golgi was observed upon ZnT4-, ZnT5/6-, ZnT7-, or ZnT4/5/6/7-silencing (Supplementary Fig. 10). Two alternative explanations seem possible for this observation; active recruitment of the COPI coatomer proteins on the Golgi membrane, or prevented detachment of COPI-coated vesicles from the Golgi under the ZnTs knockdown conditions. In either case, the above results suggest that ZnT-knockdown does not impair recruitment of COPI coatomer proteins.

To investigate whether the relocation of ERp44 caused by ZnT4-silencing depends on its $Zn^{2+}$-binding ability, we next observed the cellular localization of YFP-ERp44(3HA), a mutant lacking $Zn^{2+}$-binding ability, under the ZnT4-silencing condition. As previously reported,

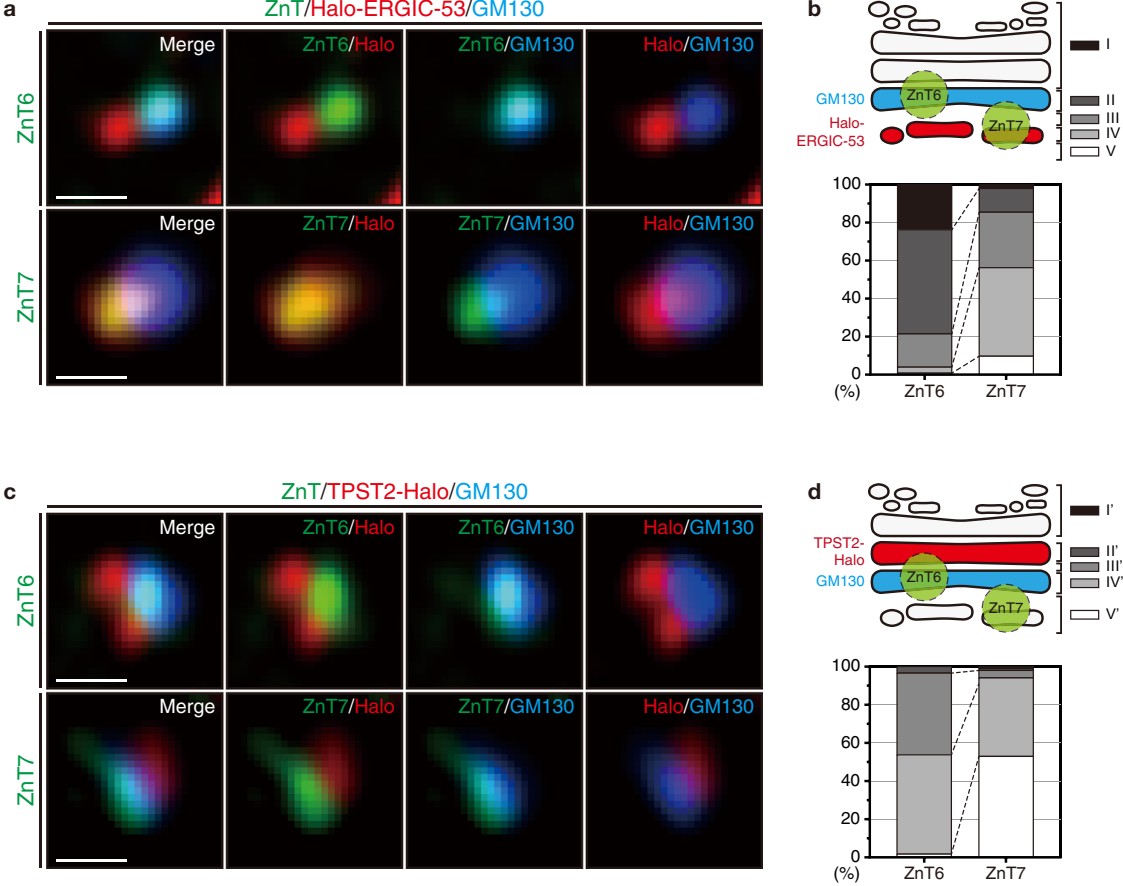

**Fig. 2 | Intracellular localization of ZnT6 and ZnT7. a** HeLa Kyoto cells expressing Halo-ERGIC-53 were labeled with 5 nM HTL-TMR for 30 min, and treated with 33 μM nocodazole for 4 h. Cells were then fixed and immunostained for GM130 and ZnT6 or ZnT7. Fluorescence images were acquired by using Airyscan super-resolution microscopy. Scale bars, 500 nm. Images are representative of three independent experiments. See also Supplementary Movies 1 and 2 for 3D projection images. **b** The relative localization of ZnT6 or ZnT7 compared to the Golgi markers, GM130 (*cis*-Golgi) and Halo-ERGIC-53 (pre-*cis*-Golgi), was classified into 5 groups as indicated. Ninety Golgi mini-stacks were counted from 6 cells in 2 independent experiments. **c** HeLa Kyoto cells expressing TPST2-Halo were treated and immunostained as in (**a**). Images are representative of three independent experiments. See also Supplementary Movies 3 and 4 for 3D projection images. **d** The relative localization of ZnT6 or ZnT7 compared to the Golgi markers, GM130 (*cis*-Golgi) and TPST2-Halo (medial Golgi), was classified into 5 groups as indicated. ≥75 Golgi mini-stacks were counted from ≥5 cells in 2 independent experiments. Source numerical data are provided in the Source Data file.

YFP-ERp44(3HA) accumulated at the Golgi[17], whereas YFP-ERp44(WT) distributed widely in the ER to Golgi area in normal cells (Supplementary Fig. 11). Notably, ZnT4-silencing reduced the Golgi localization of YFP-ERp44(WT), but not of YFP-ERp44(3HA), indicating that the enhanced ER localization of ERp44 upon ZnT4-silencing relies on its $Zn^{2+}$-binding ability (Supplementary Fig. 11). Taken together, the increased $[Zn^{2+}]_{pre\text{-}cis}$ is the most likely cause for the ERp44 relocation to the ER in ZnT4-silencing cells.

To further explore the functional roles of each ZnT family member, we performed knockdown/rescue experiments, where cells silenced for all four ZnTs were transfected with siRNA-resistant ZnT4-FLAG, PA-ZnT5 + ZnT6-FLAG, or PA-ZnT7. The siRNA resistance of these genes was confirmed by immunoblotting (Supplementary Fig. 3C). Expectedly, PA-ZnT5, ZnT6-FLAG, and PA-ZnT7 were confirmed to localize in the Golgi area (Fig. 3c). In contrast, ZnT4-FLAG displayed puncta-like signals (Fig. 3c). To confirm the exact localization of ZnT4-FLAG, we visualized ZnT4-FLAG with several organelle markers including ER, ERGIC, *cis*-Golgi, TGN, COPI, recycling endosome, early endosome, lysosome, and autophagosome. Consequently, we found that while a small portion of ZnT4-FLAG was localized at the Golgi and plasma membrane, the major portion was at the early endosomes, recycling endosomes, and lysosomes, indicating that exogenously expressed ZnT4-FLAG populates the endocytic lysosomal pathway (Supplementary Fig. 12A, B). The expression of ZnT7

efficiently blocked the aberrant Golgi accumulation of ERp44 seen in ZnT4/5/6/7-knockdown cells (Fig. 3c, d). The expression of ZnT5 + ZnT6 also showed similar effect, but that of ZnT4 did not (Fig. 3c, d). Reduced Golgi localization of ERp44 was also observed when ZnT5 + ZnT6 or ZnT7 was overexpressed in otherwise untreated cells (Supplementary Fig. 13). In contrast, overexpression of ZnT4 did not alter the Golgi-localization of ERp44 (Supplementary Fig. 13). Thus, when overexpressed, ZnT7 and ZnT5/6, but not ZnT4, can reduce the Golgi localization of ERp44 likely through the increase of the $[Zn^{2+}]$ in proximal Golgi areas.

## $Zn^{2+}$ transport activity of ZnT7 is essential for the regulation of subcellular localization of ERp44

Next, we investigated whether the $Zn^{2+}$ transport activity of ZnT7 is essential for regulating ERp44 localization. His and Asp residues on transmembrane helices 2 (TM2) and 5 (TM5) are predicted to coordinate $Zn^{2+}$ via the HDHD motif, hence indispensable for the $Zn^{2+}$ transport activity of ZnT[4]. We thus mutated His70 (on TM2) and Asp244 (on TM5) of ZnT7 to alanine, and examined whether the resultant mutant (ADHA, Fig. 4a) retained the ability to regulate the subcellular localization of ERp44. Despite the similar expression levels and intracellular localization of wild-type and mutant ZnT (Fig. 4b and Supplementary Fig. S3D), PA-ZnT7(ADHA) did not rescue the Golgi accumulation of ERp44 in ZnT4/5/6/7-knockdown cells (Fig. 4c).

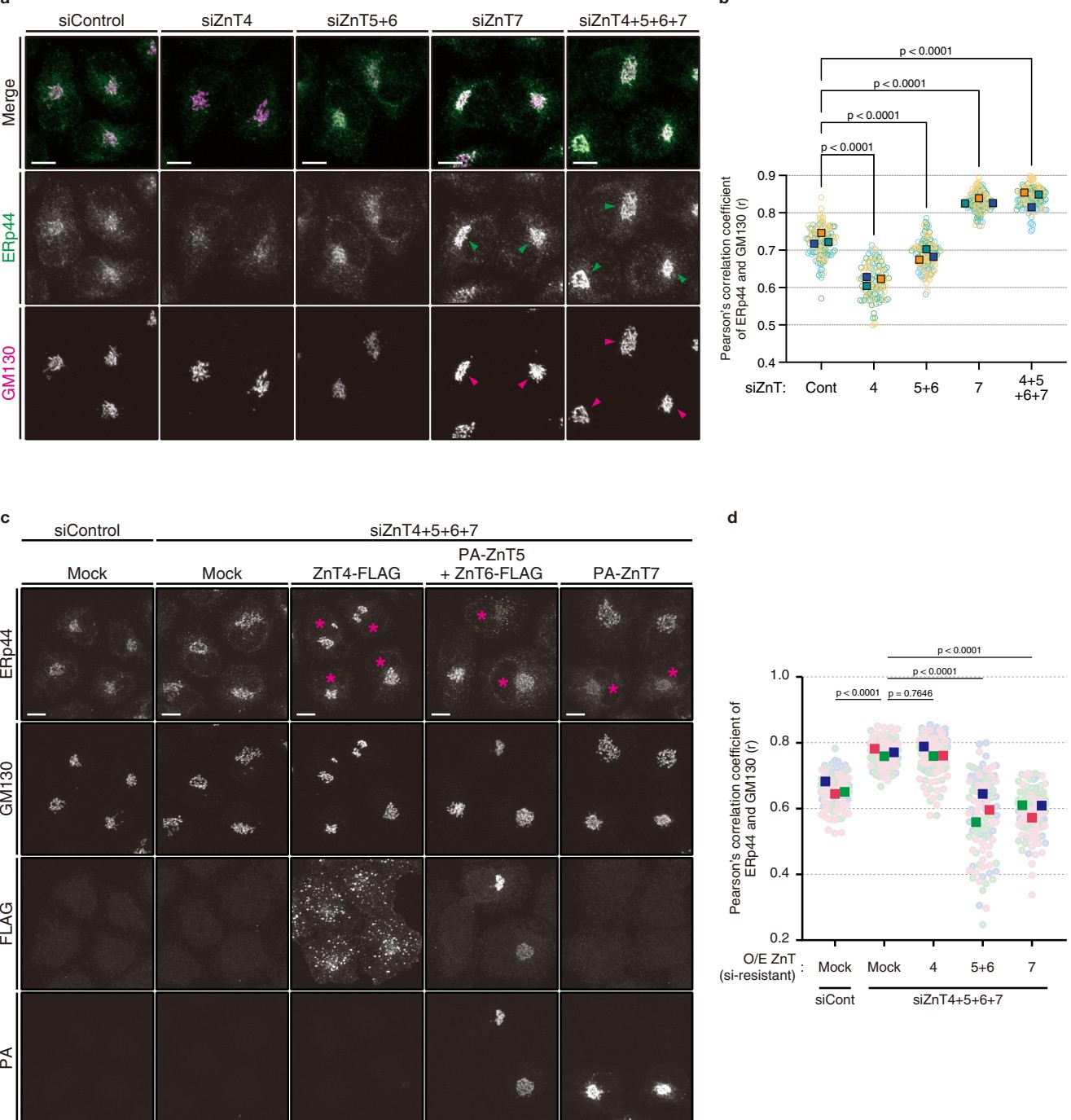

**Fig. 3 | Different contributions of the Golgi-resident ZnTs in regulating the intracellular localization of ERp44. a** Representative images of ERp44 (green) and GM130 (magenta) in HeLa Kyoto cells transfected with indicated siRNAs. Arrowheads indicate the accumulation of ERp44 in the Golgi area. Scale bars, 10 μm. **b** Quantitative analysis of Pearson's correlation coefficients of the co-localization of endogenous ERp44 and GM130 based on the immunofluorescence images shown in (**a**). Circular dots indicate each datapoint, and rectangle dots indicate the median of each experiment. Data from 3 independent experiments (siCont, *n* = 118 cells; siZnT4, *n* = 120 cells; siZnT5 + 6, *n* = 119 cells; siZnT7, *n* = 124 cells; siZnT4 + 5 + 6 + 7, *n* = 116 cells) were statistically analyzed by one-way ANOVA followed by Dunnett's multiple comparison test. **c** Knockdown/rescue experiments for ZnT4, ZnT5/6, or ZnT7. HeLa Kyoto cells pretreated with siRNAs

against ZnT4, ZnT5, ZnT6, and ZnT7 were transfected with siRNA-resistant (si-resistant) ZnT4-FLAG, PA-ZnT5 and ZnT6-FLAG, or PA-ZnT7. Cells were then fixed and immunostained for ERp44, GM130, FLAG, and PA. Magenta asterisks indicate exogenously expressed ZnT-positive cells. Scale bars, 10 μm. **d** Quantitative analysis of Pearson's correlation coefficients of the co-localization of endogenous ERp44 and GM130 based on the immunofluorescence images shown in (**c**). Circular dots indicate each datapoint, and rectangle dots indicate the median of each experiment. Data from 3 independent experiments (siCont, *n* = 119 cells; siZnT4, *n* = 119 cells; siZnT5 + 6, *n* = 132 cells; siZnT7, *n* = 109 cells; siZnT4 + 5 + 6 + 7, *n* = 120 cells) were statistically analyzed by one-way ANOVA followed by Tukey's multiple comparison test. Source numerical data are provided in the Source Data file.

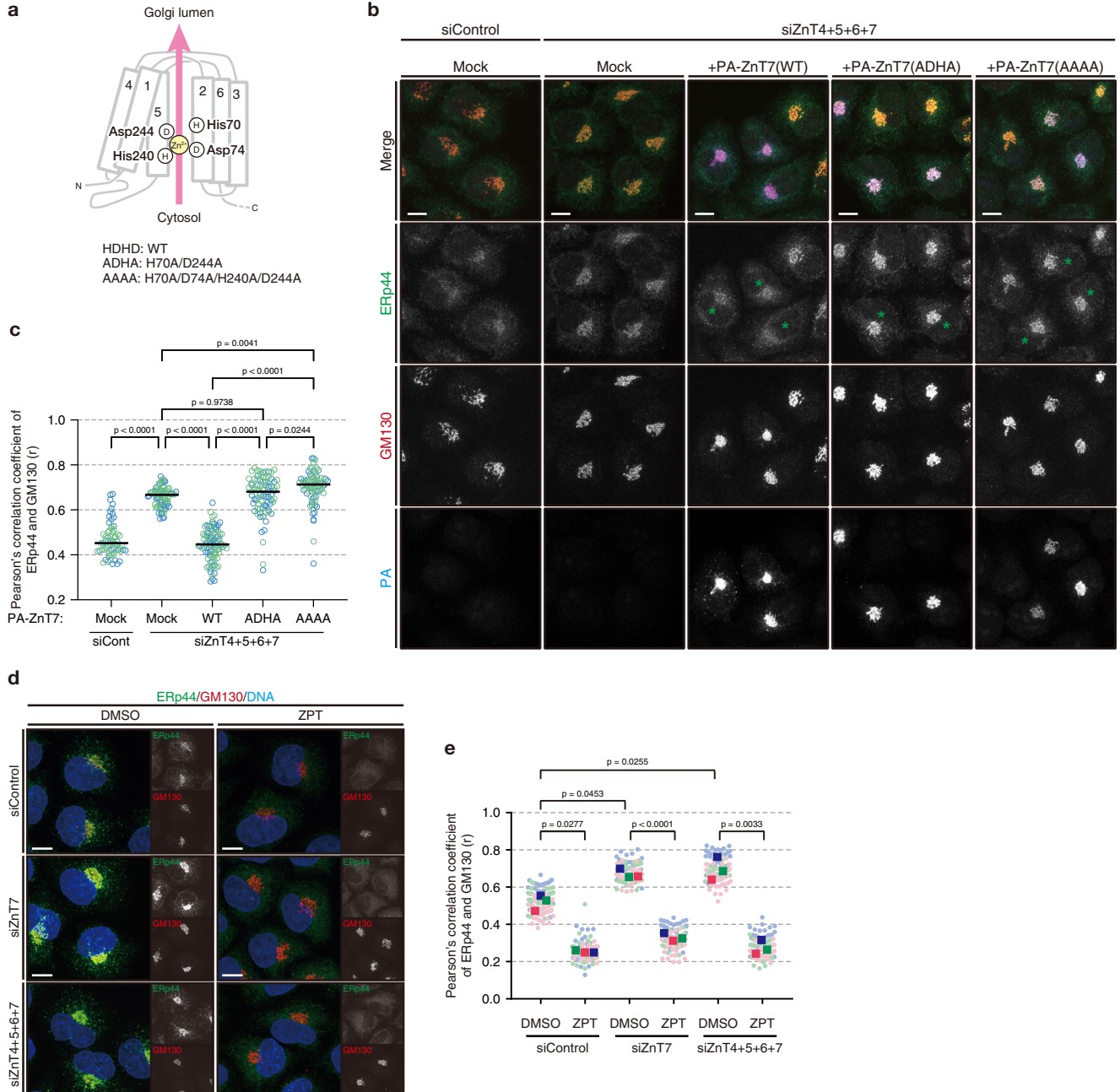

**Fig. 4 | Zn²⁺ transport activity of ZnT7 is required for the control of subcellular localization of ERp44. a** Schematic model of the ZnT7 membrane topology. His and Asp residues on TM2 and TM5 involved in Zn²⁺ coordination are indicated. **b** Immunofluorescence images showing the intracellular localization of ERp44, GM130, and PA-ZnT7. HeLa Kyoto cells were transfected with siRNAs against ZnT4, ZnT5, ZnT6, and ZnT7 simultaneously or control siRNAs. After 24 h incubation, cells were further transfected with PA-tagged siRNA-resistant ZnT7(WT), (ADHA), or (AAAA) and cultured for an additional 24 h. Cells were fixed and immunostained for ERp44 (green), GM130 (red), and PA (blue). Green asterisks indicate PA-ZnT7-positive cells. Scale bars, 10 μm. **c** Quantitative analysis of Pearson's correlation coefficients of the co-localization of endogenous ERp44 with GM130 based on the immunofluorescence images shown in (**b**). Only PA-positive cells were analyzed. Dots indicate individual data points obtained from 2 independent experiments

(siCont + Mock, *n* = 64 cells; siZnT4 + 5 + 6 + 7 + Mock, *n* = 76 cells; siZnT4 + 5 + 6 + 7 + WT, *n* = 88 cells; siZnT4 + 5 + 6 + 7 + ADHA, *n* = 82 cells; siZnT4 + 5 + 6 + 7 + AAAA, *n* = 79 cells). One-way ANOVA followed by Tukey's test was used for statistical analysis. Bars indicate the mean. **d** Effects of ZPT treatment in ZnT knockdown cells. HeLa Kyoto cells transfected with indicated siRNAs were treated with DMSO or 2.5 μM ZPT for 15 min and immunostained for ERp44 (green) and GM130 (red). Scale bars, 10 μm. **e** Quantitative analysis of Pearson's correlation coefficients of ERp44 and GM130 based on the images shown in (**d**). Circular dots indicate each datapoint, and rectangle dots indicate the median of each experiment (*N* = 3 biological replicates). One-way ANOVA followed by Tukey's multiple comparison test was used for statistical analysis. Source numerical data are provided in the Source Data file.

Similarly, the complete mutations of the motif to alanine (AAAA) impaired the rescuing ability of ZnT7 (Fig. 4b, c). Thus, the Zn²⁺ transport activity of ZnT7 is essential for the regulation of intracellular localization of ERp44.

ZnT7 was shown to activate TNAP by providing Zn²⁺ through the di-proline (PP)-motif located at the luminal loop 2[35]. To investigate whether the PP-motif is also required for the regulation of ERp44, we replaced this motif with di-alanine. The resultant mutant ZnT7 (PP-AA)

could rescue the phenotype of the ZnT4/5/6/7-knockdown at almost the same level as the wild type, suggesting that the PP-motif is not responsible for regulating the intracellular localization of ERp44 (Supplementary Fig. 3E and 14).

Importantly, $Zn^{2+}$ supply with a specific ionophore compound ($Zn^{2+}$/Pyrithione, ZPT) was able to redistribute ERp44 from the Golgi to the ER in cells lacking either only ZnT7 or all four transporters ZnT4/5/6/7 (Fig. 4d, e). Thus, we conclude that $Zn^{2+}$ per se is the primary determinant of the localization of ERp44 in the ESP.

**Golgi-resident ZnTs coordinately regulate the function of ERp44**

Next, we examined the effects of the Golgi-resident ZnT knockdown on ERp44 function. As $Zn^{2+}$-binding enhances the client-binding affinity of ERp44[17], $Zn^{2+}$ availability would likely affect the formation of ERp44-client complexes in cells. Indeed, ERp44-involving disulfide-linked oligomers formed in normal condition were greatly diminished by treating cells with TPEN, or silencing ZnT4/5/6/7 in combination (Fig. 5a). Similarly, the single knockdown of ZnT7, which reduces $[Zn^{2+}]_{pre\text{-}cis}$ (Fig. 1d), markedly decreased the formation of complexes (Fig. 5a, b). ZnT4-knockdown also inhibited the complex formation, but to a lesser extent than ZnT7-knockdown (Fig. 5a, b). This observation can be explained by the increased $[Zn^{2+}]_{pre\text{-}cis}$ upon ZnT4-knockdown (Fig. 1d). Presumably, the increased $[Zn^{2+}]_{pre\text{-}cis}$ promotes ERp44 to return to the ER (Fig. 3a) and release clients there due to the low $[Zn^{2+}]_{ER}$[19]. The ZnT5/6-knockdown displayed marginal effects on the formation of the ERp44-client complexes although it greatly reduced $[Zn^{2+}]$ in the medial/trans-Golgi (Fig. 5a, b). These data suggest that ERp44 may form disulfide-linked oligomers primarily in the pre-cis-Golgi, and that $Zn^{2+}$ in the distal Golgi areas affects the formation or accumulation of the oligomers to smaller extent than that in the proximal Golgi areas.

We previously showed that $Zn^{2+}$-chelation with TPEN impaired the intracellular retention of ERp44[17]. We therefore investigated whether the knockdown of ZnTs allows ERp44 secretion. For this purpose, ZnTs knockdown cells were transfected with Halo-ERp44, and the cell lysates and conditioned media were analyzed by immunoblotting. Notably, the secretion of exogenously expressed Halo-ERp44 was significantly increased upon silencing of ZnT7 or ZnT4/5/6/7 (Fig. 5c, d), consistent with the enhanced Golgi accumulation of ERp44 under this condition (Fig. 3a). By contrast, ZnT4- or ZnT5/6-knockdown only marginally affected ERp44 secretion (Fig. 5c, d), suggesting different roles of the Golgi-resident ZnTs.

Next, the client retrieval activity of ERp44 was assessed under various ZnTs knockdown conditions by analyzing the secretion of its client proteins Ero1α, ERAP1 and Prx4 (Fig. 5e–j)[11,13,15,36]. A redox-inactive Ero1α mutant, with Cys99 and Cys104 replaced by Ala, was used to prevent undesired perturbation of ESP redox homeostasis (i.e., hyper-oxidation). Consistent with our previous report[17], simultaneous knockdowns of ZnT4, 5, 6, and 7 increased the secretion of all three clients (Fig. 5e–j). Whereas neither ERAP1 nor Ero1α was secreted efficiently by cells with lower ZnT7 levels, ZnT4- or ZnT5/6-knockdown significantly enhanced their secretion (Fig. 5e–h), implying a critical role of ZnT4 and ZnT5/6 in the ERp44-dependent client retrieval. Similar observation was made when employing Prx4 as a client, whilst ZnT7-knockdown slightly enhanced its secretion (Fig. 5i, j). Thus, ZnT4/5/6 appear to have greater responsibility for the ERp44-mediated client retrieval cycle than ZnT7. One likely explanation is that ZnT4/5/6, which import $Zn^{2+}$ into the distal Golgi area (Figs. 1 and 2), serve to guarantee ERp44 to capture and retrieve clients throughout the Golgi, while under normal conditions, ERp44 does so at the pre-cis-Golgi in a ZnT7-dependent manner. In other words, ZnT4/5/6-mediated $Zn^{2+}$ in the distal Golgi area plays a gatekeeper role for ERp44 to work as an effective inhibitor of undesired protein secretion (see also Discussion).

**Golgi-resident ZnTs finely tune the traffic of ERp44**

To gain deeper insight into the $Zn^{2+}$-regulated traffic and client retrieval cycle of ERp44, we set up a retention using selective hooks (RUSH) system[37]. To this end, we extended ERp44 at its N-terminus with a streptavidin binding peptide (SBP) and a FLAG tag, while streptavidin-KDEL was used as an ER-retention hook (Fig. 6a). As expected, SBP-FLAG-ERp44 was mainly localized in the ER (Fig. 6b). Upon biotin addition, SBP-FLAG-ERp44 started relocating from the ER to the Golgi, and this relocation was completed within 45 min. Thereafter, the intensity of the SBP-FLAG-ERp44 signals on the Golgi was decreased (Fig. 6b). A time-lapse imaging combined with the RUSH system further verified the biotin-triggered ER-to-Golgi relocation of SBP-Halo-ERp44 (Fig. 7). After reaching the Golgi, SBP-Halo-ERp44(WT) was seemingly kept localized around the Golgi and its peripheral wide area with network structures during the observation period of 60 min (Supplementary Fig. 15 and Supplementary Movie 5), suggesting that SBP-Halo-ERp44(WT) is cycling between the ER and Golgi. Similarly, a mutant lacking the RDEL ER-retrieval signal, SBP-Halo-ERp44(ΔRDEL), exhibited the biotin-triggered ER-to-Golgi relocation. However, its Golgi localization was greatly reduced at 40–60 min after the biotin stimulation, probably due to the exit from the Golgi to the downstream compartments (Supplementary Fig. 15 and Supplementary Movie 6). Thus, our RUSH systems visualized the synchronized anterograde transport of ERp44 in the ESP and the different destinations of ERp44 (WT) and ERp44 (ΔRDEL) after reaching the Golgi.

Quantitative analyses of the relative signal intensity of SBP-FLAG-ERp44 on the Golgi ($F_{Golgi/Total}$) demonstrated that the knockdowns of ZnT4, 5/6, and 7 affected the kinetics and behavior of ERp44 traffic in different manners (Fig. 6c and Supplementary Fig. 16A). ZnT4-knockdown cells showed only small increment of $F_{Golgi/Total}$ and reached a plateau already at 15 min after biotin addition. ZnT5/6-knockdown cells displayed a continual increase of $F_{Golgi/Total}$ over 60 min, but slower than siControl cells. In contrast, both ZnT7- and ZnT4/5/6/7-silenced cells showed a similar initial increment of $F_{Golgi/Total}$ to siControl cells, but no significant decrease in $F_{Golgi/Total}$ during 45–60 min.

To observe the ER-to-Golgi anterograde transport of ERp44 exclusively and thereby interpret the kinetics of the ERp44 traffic more precisely, we also quantitatively analyzed the biotin-triggered relocation of SBP-FLAG-ERp44(ΔRDEL) (Fig. 6d and Supplementary Fig. 16B). In siControl cells, the $F_{Golgi/Total}$ of SBP-FLAG-ERp44(ΔRDEL) increased up to 15–30 min and then decreased, likely reflecting its transport to the downstream compartments and extracellular space. In support of this, significant amount of SBP-FLAG-ERp44(ΔRDEL) was secreted out to the medium (Supplementary Fig. 16C). Intriguingly, this mutant displayed more rapid increment of $F_{Golgi/Total}$ up to 30 min in ZnT4-silencing cells (Fig. 6d), indicating that the ER-to-Golgi transport itself was not impaired by the knockdown of ZnT4. Nevertheless, the $F_{Golgi/Total}$ of SBP-FLAG-ERp44(WT) increased only marginally under this condition (Fig. 6c). Considering that the $[Zn^{2+}]_{pre\text{-}cis}$ was abnormally elevated in ZnT4-knockdown cells (see Fig. 1c), we surmise that the retrograde transport of ERp44 from the Golgi to the ER is accelerated in ZnT4-silencing cells. On the other hand, the ZnT5/6 knockdown showed no significant changes in $F_{Golgi/Total}$ of ERp44(ΔRDEL) after the biotin trigger. Given that greater amounts of ERp44(ΔRDEL) were secreted out by ZnT5/6-silencing cells (Supplementary Fig. 16C), it can be interpreted that the anterograde intra-Golgi trafficking of ERp44 is accelerated under this condition so that the rates of the Golgi entry and exit of ERp44(ΔRDEL) are almost equal to each other. In ZnT7-knockdown cells, similar to control cells, the $F_{Golgi/Total}$ of ERp44(ΔRDEL) was slightly but significantly increased at 15 and 30 min, and then decreased gradually during 30–60 min to reach almost the same level as at 0 min. ZnT4/5/6/7-silenced cells also exhibited an increment of $F_{Golgi/Total}$ at 15 min, but significantly decreased thereafter, suggesting

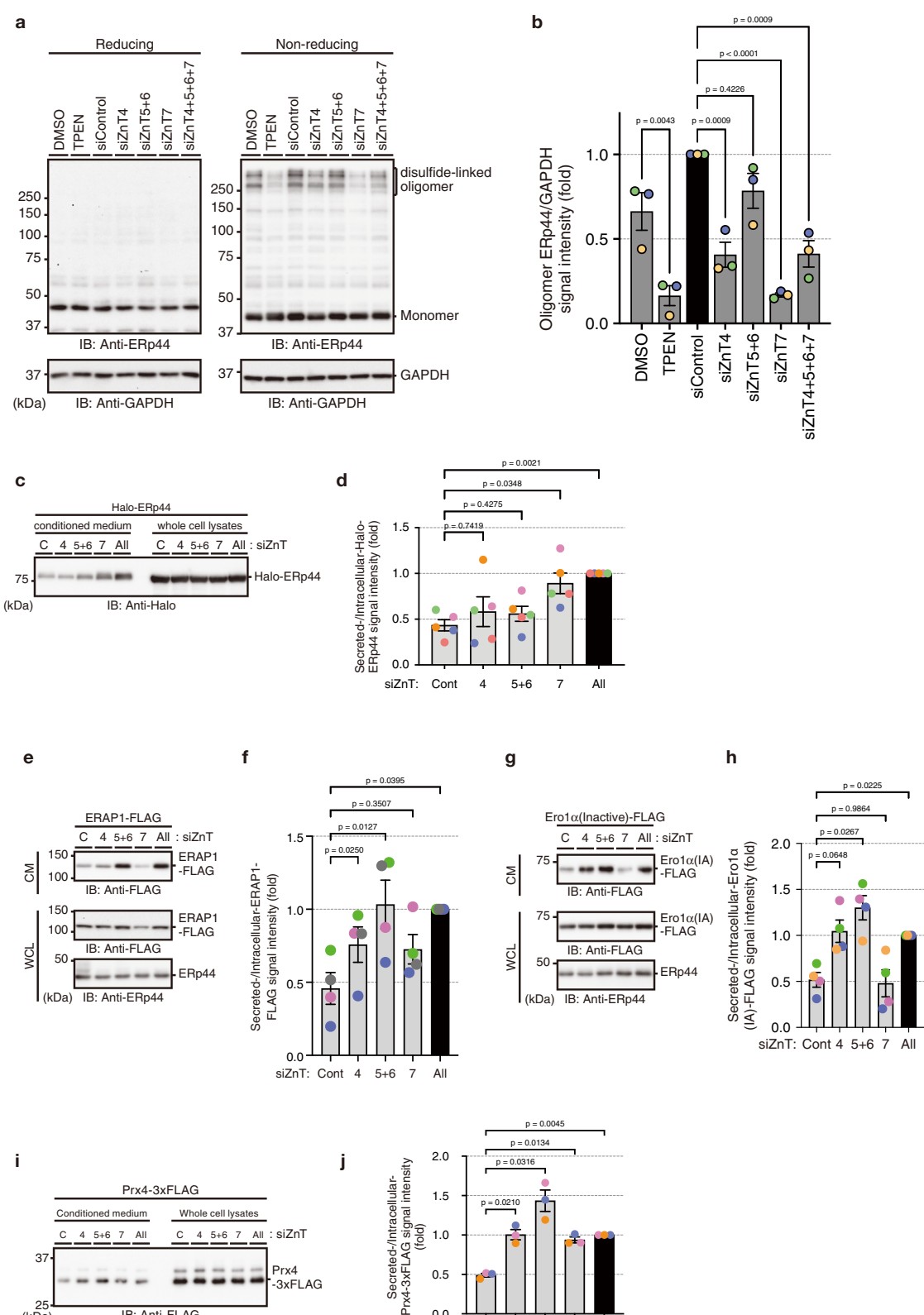

that under all ZnTs knockdown condition, ERp44(ΔRDEL) can proceed on the secretory pathway as in normal cells. These results suggest that ZnT7- or ZnT4/5/6/7-knockdown does not affect the ER-to-Golgi transport of ERp44 per se, consistent with the similar level of secretion of ERp44(ΔRDEL) between siControl and siZnT7 or siZnT4/5/6/7 cells (Supplementary Fig. 16C). The prolonged Golgi-

accumulation of ERp44(WT) in the ZnT7- or ZnT4/5/6/7-knockdown cells is likely caused by prevention of the Golgi-to-ER backward trafficking due to the decreased [Zn²⁺]_pre-cis. Thus, ERp44 cycling was considerably influenced by ZnTs knockdowns, reinforcing the importance of ZnTs-mediated Zn²⁺ homeostasis in ERp44-mediated protein quality control.

**Fig. 5 | Different effects of the Golgi-resident ZnTs knockdown on ERp44 function. a** Whole-cell lysates that had been precipitated with TCA and treated with NEM were resolved by SDS-PAGE and analyzed by immunoblotting with anti-ERp44 and anti-GAPDH antibodies under reducing (left) and non-reducing (right) conditions. **b** The signal intensity of disulfide-linked oligomers involving ERp44 relative to that of GAPDH was quantified. Data are the means ± SEM ($N$ = 3 biological replicates). Dots indicate the individual data points. One-way ANOVA followed by Dunnett's test was used for statistical analysis. **c, d** Effects of ZnT-silencing on the intracellular retention of ERp44. HeLa Kyoto cells pretreated with siRNAs against indicated ZnT members were transfected with Halo-ERp44. After 36 h incubation, cells were washed twice and incubated in Opti-MEM for additional 6 h. TCA-precipitated conditioned media and whole cell lysates were resolved by reducing SDS-PAGE and analyzed by immunoblotting using anti-Halo antibody. The signal intensity of Halo-ERp44 secreted out of cells relative to that of whole cell lysates was quantified and shown in (**d**). Data are the means ± SEM ($N$ = 5 biological replicates). Dots indicate the individual data points. One-way ANOVA followed by Dunnett's test was used for statistical analysis. **e–j** Effects of ZnT-silencing on the client-retention activity of ERp44. HeLa Kyoto cells pretreated with siRNAs against indicated ZnT members were transfected with ERAP1-FLAG (**e**), Ero1α (inactive)-FLAG (**g**) or Prx4-3×FLAG (**i**). Secreted ERAP1-FLAG or Ero1α (inactive)-FLAG was harvested from conditioned media (CM) by immunoprecipitation using anti-DYKDDDDK antibody agarose beads. Secreted Prx4-3×FLAG was concentrated by TCA precipitation. Precipitants and whole cell lysates (WCL) were analyzed by immunoblotting. The relative levels of secreted ERAP1-FLAG, Ero1α (inactive)-FLAG and Prx4-3×FLAG quantified by densitometry are shown in (**f**), (**h**), and (**j**), respectively. Data are the means ± SEM ($N$ = 4 biological replicates for (**f**) and (**h**), and $N$ = 3 biological replicates for (**j**)). Dots indicate the individual data points. One-way ANOVA followed by Dunnett's test was used for statistical analysis. Source numerical data and unprocessed blotting images are provided in the Source Data file.

## Knockdowns of Golgi-resident ZnTs do not impair ER-to-Golgi trafficking

To gain more general insight into roles of $Zn^{2+}$ in the ER-to-Golgi protein trafficking, we employed another model cargo protein in which the TM domain of ST6 β-galactoside sialyltransferase (ST), one of the Golgi-resident membranous proteins, was fused to the N-terminus of SBP-EGFP (Fig. 7a), and applied it to the RUSH experiments under the ZnTs knockdown conditions. ZnT(s)-silencing cells were co-transfected with pRUSH-ST-SBP-EGFP and Halo-ManII, a Golgi-marker. Cells were labeled with HTL-TMR and subjected to time-lapse imaging. Expectedly, ST-SBP-EGFP was initially localized at the ER-like area, and relocated to the Golgi upon the biotin stimulation (Fig. 7b). We quantified the intensities of EGFP and TMR fluorescent signals on the Golgi area, $GFP_{Golgi}$ and $TMR_{Golgi}$, respectively, and calculated their ratio ($GFP_{Golgi}/TMR_{Golgi}$). The $t_{1/2}$ value was determined by a curve fitting for the time range of the ratio increase (Fig. 7c), and used to assess the effects of ZnTs-knockdown on the trafficking of ST-SBP-EGFP. Control cells showed a $t_{1/2}$ value of 10.2 ± 2.2 min, and this value was almost unchanged by any ZnTs-silencing (Fig. 7d). In conclusion, the ER-to-Golgi trafficking was not substantially affected by $[Zn^{2+}]$ changes caused by the knockdown of ZnT4, ZnT5/6, ZnT7, or ZnT4/5/6/7.

For comparison, we also investigated the effects of ZnTs-silencing on the trafficking of SBP-Halo-ERp44 using the RUSH system. The live-cell imaging of the ER-to-Golgi transport of SBP-Halo-ERp44 labeled with HTL-TMR (Fig. 7f–i, Supplementary Movies 7 and 8) indicated that the trafficking of SBP-Halo-ERp44 was ~10-fold faster ($t_{1/2}$ = 0.92 ± 0.71 min) than that of ST-SBP-EGFP ($t_{1/2}$ = 10.2 ± 2.2 min) in siControl cells, suggesting that cells have a system that accelerates the ER-to-Golgi transport of ERp44, as previously reported[38]. In support of the immunofluorescence results using the fixed cells (Fig. 6c), ZnT4-silencing did not allow SBP-Halo-ERp44 to relocate to the Golgi (Fig.7f, Supplementary Movie 8). This observation is consistent with our notion that in ZnT4-silencing cells with higher $[Zn^{2+}]_{pre\text{-}cis}$ (Fig. 1d), ERp44 is transported back to the ER immediately after reaching this compartment. Of note, ZnT7- or ZnT4/5/6/7-silencing allowed the relocation of SBP-Halo-ERp44 to the Golgi, but significantly increased the $t_{1/2}$ value (Fig. 7h), whereas ZnT5/6-silencing did not (Fig. 7h). The initial slope of the $TMR_{Golgi}$ signal increment ($v_0$) (Fig. 7g), which represents the rate of entry to the Golgi of SBP-Halo-ERp44, was not largely altered by the ZnTs knockdowns (Fig. 7i). Thus, $[Zn^{2+}]_{Golgi}$ seems unlikely to affect the ER-to-Golgi transport of ERp44 per se. We presume that lowering $[Zn^{2+}]_{Golgi}$ facilitates the secretion of ERp44 from the Golgi to the downstream compartments (Fig. 5c), leading to the apparently slower increase of the signals derived from Golgi-reached ERp44. As a result, it took a longer time until the signals reached a maximum plateau under the ZnT7- or ZnT4/5/6/7-silencing condition.

## Discussion

$Zn^{2+}$ import into the ESP is essential for many key ectoenzymes[35,38–43]. By determining the localization of ZnT complexes and quantifying $[Zn^{2+}]$ in different Golgi cisternae, this study provides important insights into how each ZnT regulates $Zn^{2+}$ homeostasis and ERp44 cycling and function in the ESP.

The first conclusion emerging from our high-resolution imaging analyses is that $[Zn^{2+}]$ changes dramatically amongst sequential ESP stations. Whilst picomolar in the ER, $[Zn^{2+}]$ reaches its maximum (~100 nM) in the most proximal Golgi cisterna (referred to herein as the pre-cis-Golgi). Lower levels (~60–80 nM) are then found in the cis-, medial- and trans-Golgi cisterna. This gradient seems likely to be a result of the discrete distribution of the three main ZnT complexes. Super-resolution microscopy analyses revealed that ZnT7 and ZnT5/ZnT6 accumulate mainly in the pre-cis-Golgi and cis-/medial-Golgi, respectively (Fig. 2a, c). The expression pattern of FLAG-tagged ZnT4 suggests that this transporter occupies distal regions of the secretory pathway, including the plasma membrane and endolysosomal pathway (Supplementary Fig. 12). On the whole, the sequential distribution of ZnT7, ZnT5/6 and ZnT4 explains the different effects of their knockdowns on the $[Zn^{2+}]_{Golgi}$. At the same time, it implies that they play distinct functions. While the localization of ZnT6 was shown to rely on its association with ZnT5[34], the mechanisms underlying the differential ZnTs distribution are poorly understood. Unlike ZnT7, ZnT5 and ZnT4 are predicted to have extra N-terminal TM domain(s) that may define their localization via interactions with specific partner proteins. Further protein engineering and proteomic approaches are necessary to elucidate what determines the traffic and destinations of ZnT4/5/6/7.

An unexpected finding was that ZnT4-silencing increased $[Zn^{2+}]_{cyto}$ and $[Zn^{2+}]_{pre\text{-}cis}$; the presence of ZnT4 in the plasma membrane might mediate $Zn^{2+}$ efflux from the cytosol to the extracellular space (Supplementary Fig. 12). Previous work demonstrated that excess cytosolic $Zn^{2+}$ can be incorporated into ERGIC-53-positive vesicles in S-nitrosocysteine-treated C6 cells[44]. Likewise, the higher $[Zn^{2+}]_{cyto}$ in our ZnT4-silencing HeLa Kyoto cells may be imported into pre-cis-Golgi via ZnT7, resulting in the higher $[Zn^{2+}]_{pre\text{-}cis}$. It is possible that the Golgi-resident ZIPs also contribute to $Zn^{2+}$ homeostasis in the Golgi. Further analyses will reveal the deeper mechanisms of how ZnTs and ZIPs cooperatively regulate the $[Zn^{2+}]_{Golgi}$.

The use of targeted pHluorin2 sensors confirmed that a rather steep pH gradient exists along the ESP. The pH value of the cis-Golgi is around 6.5, whereas the ER and pre-cis-Golgi maintain a neutral pH (Supplementary Fig. 2F). Such $[Zn^{2+}]$ and pH gradients are key in regulating functions of ERp44, whose affinity for $Zn^{2+}$ ranges from 135 nM at pH 7.2 to 295 nM at pH 6.2[17]. The Golgi-to-ER retrieval of ERp44 depends on KDELRs that bind the C-terminal RDEL motif of ERp44[15], mainly in pre-cis-/cis-Golgi[22,23,45,46]. Given that the affinity of KDELRs for its clients increases at weakly acidic

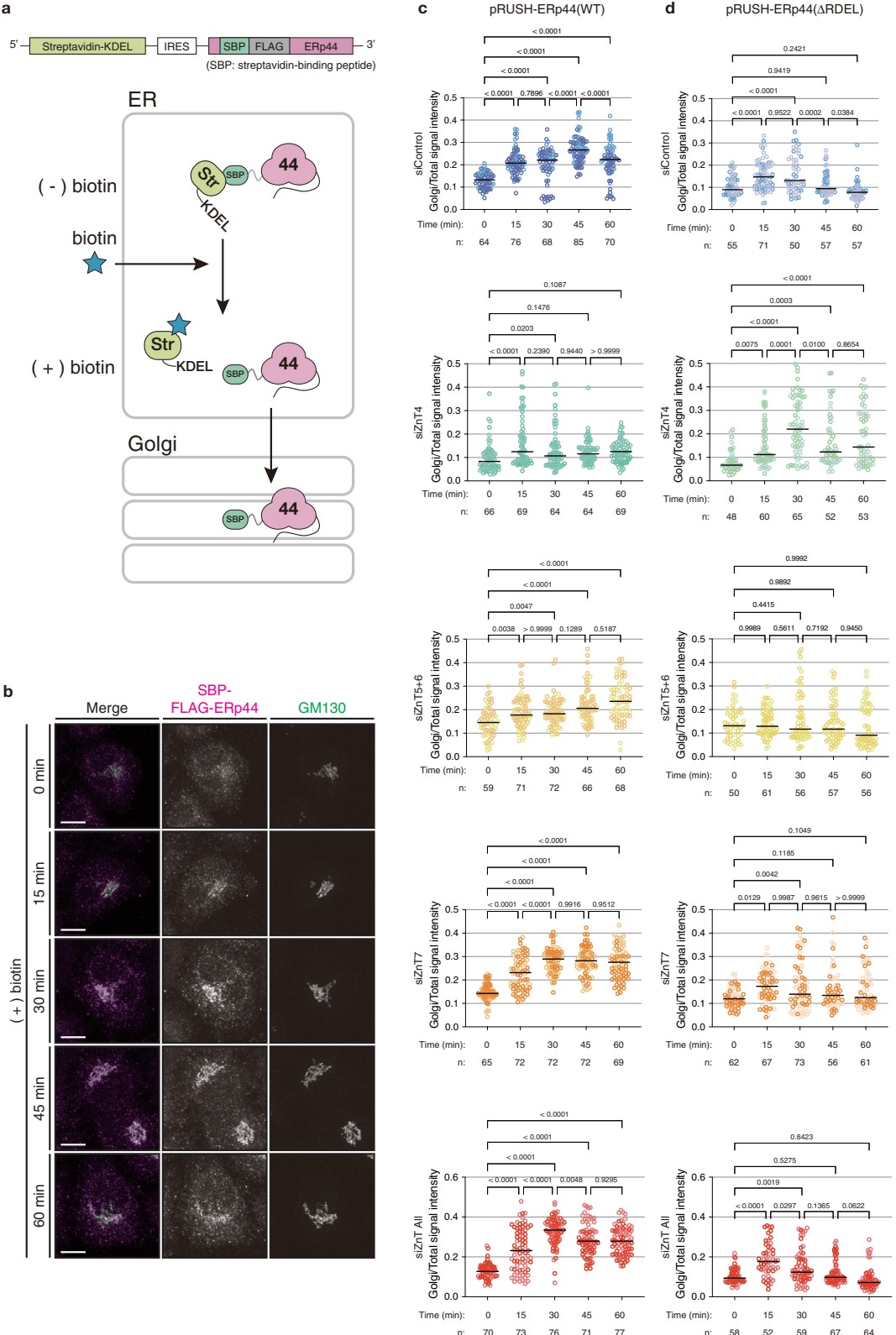

**Fig. 6 | ZnT controls the kinetics of ERp44 trafficking between the ER and Golgi.**
**a** Scheme of the pRUSH-ERp44 constructs, and of the principle of the assay. Streptavidin (Str)-KDEL and SBP-FLAG-ERp44 are separately and successively expressed by virtue of the insertion of an internal ribosome entry site (IRES) between these two. SBP-FLAG-ERp44 is retained in the ER via its interaction with Str-KDEL. By addition of biotin, SBP-FLAG-ERp44 is released and transported from the ER to the Golgi in a synchronized manner. **b** Representative images of SBP-FLAG-ERp44 localization before and after biotin treatment. Scale bars, 10 μm.

**c**, **d** Quantitative analyses of relative signal intensity of SBP-FLAG-ERp44 (WT) (**c**) or SBP-FLAG-ERp44 (ΔRDEL) (**d**) on the Golgi apparatus in cells transfected with indicated siRNAs. Dots indicate individual data points, and the sample size (n) indicates the number of cells from two independent experiments. Data obtained from different experiments are shown in different color. Bars indicate the median. One-way ANOVA followed by Tukey's test was used for statistical analysis. Source numerical data are provided in the Source Data file.

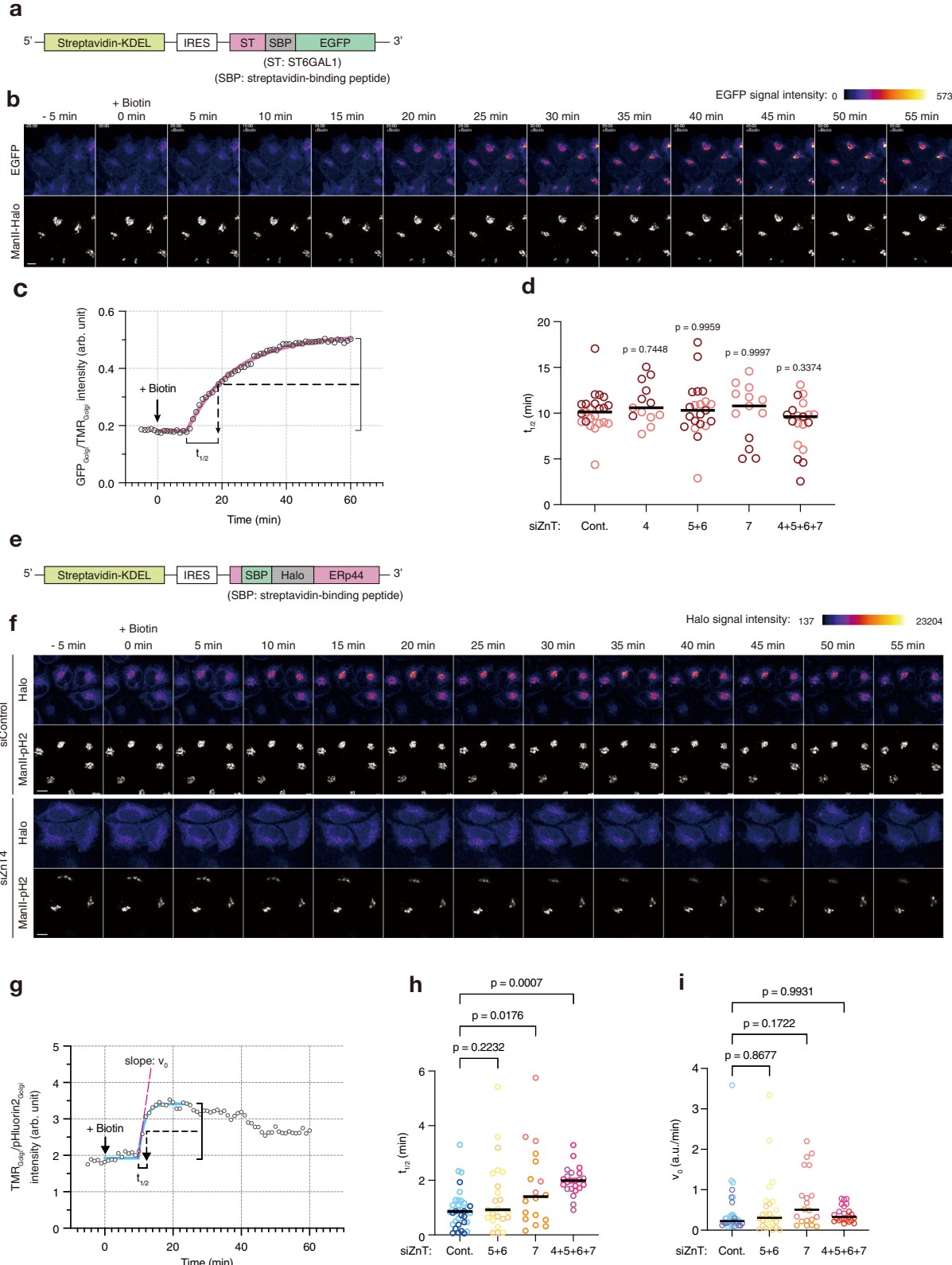

pH[47,48], KDELR most likely initiates the retrograde transport of ERp44 at the *cis*-Golgi.

Taken all together, our findings allow us to propose a refined model of protein quality control in the ESP, ensured by the cooperation of ERp44 and KDELR, modulated by the [$Zn^{2+}$] and pH gradients (Fig. 8), and maintained by the sequential distribution of ZnT7, 5/6 and 4 complexes in post-ER compartments. ERp44 acquires $Zn^{2+}$ and forms complexes with its clients or $Zn^{2+}$-bridged homodimers[17] as it enters the pre-*cis*-Golgi, where [$Zn^{2+}$] is maximal in the ESP. When the client-ERp44 complexes reach the weakly acidic *cis*-Golgi, KDELR captures and retrieves the complexes to the ER via COPI-coated vesicles. The complexes are disassembled in the ER, owing to the low [$Zn^{2+}$] and neutral pH in this organelle[18]. Once client proteins are released from ERp44, they become functional as ER-resident enzymes, or they

**Fig. 7 | Time-lapse imaging of the ER-to-Golgi trafficking of ERp44 under various ZnT knockdown conditions. a** Scheme of the pRUSH-ST-SBP-EGFP construct. **b** Representative snapshots of the time-lapse imaging in the RUSH assay of ST-SBP-EGFP. Fluorescence images were acquired every 1 min. Upper and lower panels indicate the signals of ST-SBP-EGFP and ManII-Halo (Golgi marker), respectively. Biotin addition time is set as 0 min. Scale bar, 10 μm. **c** Ratios of intensities of GFP and TMR signals were plotted as a function of observation time (black circles). A fitting curve was calculated from the plots and indicated as a purple line. The $t_{1/2}$ value was calculated based on the fitting curve (see Methods for more details). **d** $t_{1/2}$ values of ST-SBP-EGFP in HeLa Kyoto cells transfected with indicated siRNAs. Circles indicate individual data points obtained from 2 independent experiments (siCont, $n = 23$ cells; siZnT4, $n = 13$ cells; siZnT5 + 6, $n = 21$ cells; siZnT7, $n = 13$ cells; siZnT4 + 5 + 6 + 7, $n = 19$ cells), and bars indicate medians. One-way ANOVA followed by Dunnett's test was used for statistical analysis. **e** Scheme of the pRUSH-

SBP-Halo-ERp44 constructs. **f** Representative snapshots of the time-lapse imaging in the RUSH assay of SBP-Halo-ERp44. Fluorescence images were acquired as in (**b**) Upper and lower panels under each condition indicate the signals of SBP-Halo-ERp44 and ManII-pHluorin2 (Golgi marker), respectively. Scale bars, 10 μm. See also Supplementary Movies 7 and 8. **g** Ratios of intensities of TMR and pHluorin2 signals were plotted as a function of observation time (black circles). A fitting curve was calculated from the plots, and indicated as a blue line. The $t_{1/2}$ and initial slope ($v_0$) values were estimated based on the fitting curve (see Methods for more details). **h**, **i** $t_{1/2}$ and $v_0$ values of SBP-Halo-ERp44. Circles indicate individual data points obtained from 2 independent experiments (siCont, $n = 34$ cells; siZnT5 + 6, $n = 24$ cells; siZnT7, $n = 18$ cells; siZnT4 + 5 + 6 + 7, $n = 23$ cells), and bars indicate the median. One-way ANOVA followed by Dunnett's test was used for statistical analysis. Source numerical data are provided in the Source Data file.

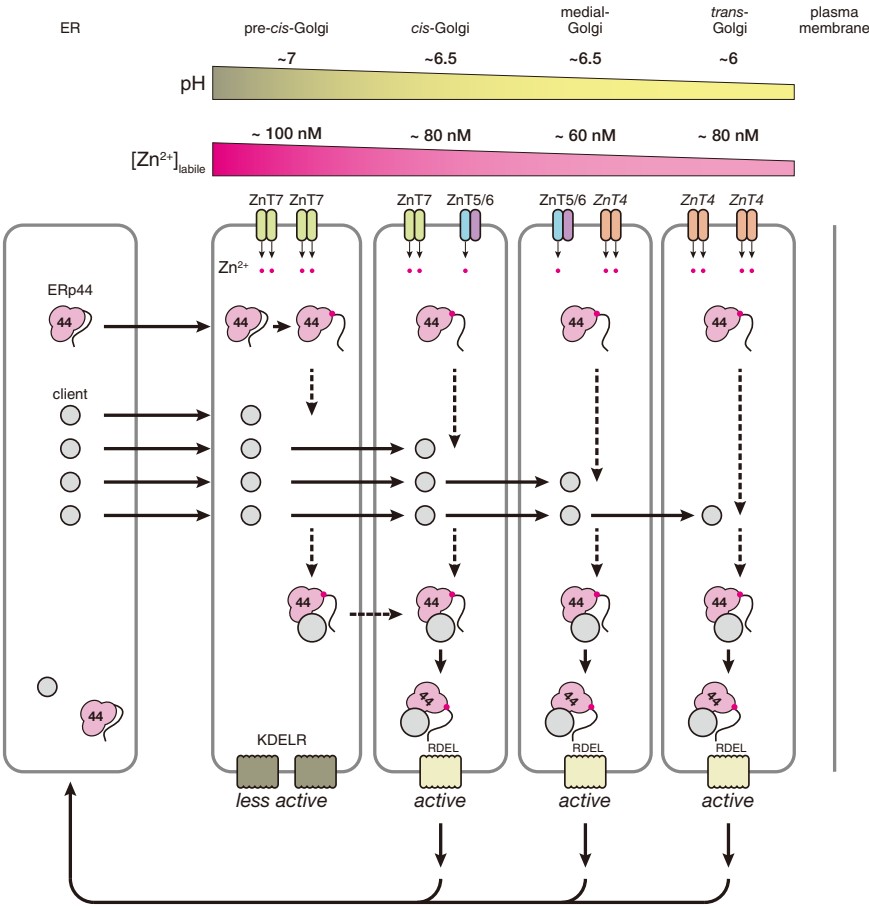

**Fig. 8 | Proposed model of ERp44 trafficking regulated by Golgi-resident ZnT members for protein quality control in the early secretory pathway.** ERp44 acquires $Zn^{2+}$ in the proximal Golgi with relatively high $[Zn^{2+}]$ and neutral pH, and forms complexes with clients (shown as grey circles) in this compartment. Most complexes are then transported to the *cis*-Golgi and recognized by KDELR, more active than in the pre-*cis*-Golgi because of the lower pH. Eventually, complexes are retrogradely transported back to the ER. ZnT7 maintains high $[Zn^{2+}]$ in the proximal Golgi and facilitates $Zn^{2+}$-metalation of ERp44, whereas ZnT4 and ZnT5/ZnT6 are important for $Zn^{2+}$ homeostasis in the medial- and *trans*-Golgi, and promote ERp44 to bind clients also in the distal Golgi[14].

resume folding and assembly to become mature proteins with the assistance of ESP chaperones. As an additional route, $Zn^{2+}$-bound ERp44 may likely be transported back to the ER even without client binding, as ZPT treatment alone was sufficient for ERp44 to exit from the Golgi (Fig. 4c, d), and only traces of ERp44 are secreted by cells overexpressing it[12,14,17].

The present study also demonstrated that ZnT7-knockdown reduced the formation of disulfide-linked complexes involving ERp44 in cells (Fig. 5a). Without bound clients, ERp44 would expose the RDEL-motif to lesser extent[9,12,14,17], preventing ERp44 retrieval from the *cis*-Golgi and allowing more ERp44 movement to the distal Golgi

areas. However, substantial amount of ERp44 remains intracellular (mostly in the Golgi) under the ZnT7-knockdown condition (Fig. 5c, e, g). Moreover, ZnT7-knockdown showed moderate effects on $[Zn^{2+}]$ in the *cis*-, medial- and *trans*-Golgi. Hence, ERp44 could acquire $Zn^{2+}$ and bind client proteins at these Golgi cisternae even after ZnT7-knockdown. Under such circumstances, the complex would be readily recognized by KDELR, whose affinity is increased by lower pH values, and rapidly transported back to the ER, where the complex would be disassembled through $Zn^{2+}$ release. In this way, the lifetime of the ERp44-client complexes can be shortened in ZnT7-knockdown cells, leading to the decrease of disulfide-linked complexes (Fig. 5a). In

contrast, since ZnT4-knockdown increases the $[Zn^{2+}]_{pre\text{-}cis}$ (Fig. 1d), more ERp44 can bind $Zn^{2+}$ and be retrieved to the ER. Consistently, less ERp44 accumulates in the Golgi of ZnT4-siliencing cells (Fig. 3a, b), accounting for their tendency to secrete ERp44 clients (Fig. 5e–j). ZnT5/6-knockdown markedly lowered $[Zn^{2+}]$ in the cis-, medial- and trans-Golgi while it little affected the $[Zn^{2+}]_{pre\text{-}cis}$ (Fig. 1d). We surmise that $Zn^{2+}$ imported by ZnT4 and ZnT5/6 in the distal Golgi areas serves as a gatekeeping factor for the ERp44-mediated client retention, and its deprivation caused by ZnT4- or ZnT5/6-knockdown would allow even greater secretion of client proteins (Fig. 5e–j).

Our previous study revealed that acidic pH in the Golgi can activate ERp44 function by increasing the flexibility of its C-terminal tail[9,12]. While the crosstalk between pH and zinc in the ESP is not fully understood, the present study demonstrates that lower ZnTs levels considerably affects ERp44 functions without significant changes in pH values at any Golgi cisternae (Supplementary Fig. 2G, H). In this regard, $Zn^{2+}$ in the ESP acts as a more dominant regulator of ERp44 than pH. Presumably, pH homeostasis in the Golgi works upstream of $Zn^{2+}$ homeostasis via stimulation of $Zn^{2+}/H^+$ antiporter activity of ZnTs, enabling the indirect modulation of ERp44 by pH.

ERp44-KO mice were reported to show growth retardation, cardiac developmental and functional defects and hypotension[13,49]. Similarly, ZnT5-KO mice show growth retardation and male-specific death because of the arrhythmias[50]. It is possible that ZnT5-KO inhibits the function of ERp44 and allows aberrant secretion of male-specific ERp44-client proteins, leading to the heart failure. Secretome analyses with ZnT5-KO cells will identify proteins that may cause the above phenotypes, and further elucidate the physiological roles of $Zn^{2+}$-mediated protein quality control in the ESP.

In summary, we demonstrated that the $[Zn^{2+}]$ in each Golgi cisterna is strictly regulated by multiple ZnT family members. Strategies applied in this work will also be useful for studies on physiological function of the ER- and Golgi-resident ZIP members, which export $Zn^{2+}$ from the organelle lumen to the cytosol and regulate the cellular $Zn^{2+}$ homeostasis in concert with ZnTs. Further extensive studies will provide deeper and more comprehensive insight into mechanisms of $Zn^{2+}$ homeostasis and its close relation to proteostasis in the ESP.

## Methods

### Cell culture and transfection
HeLa Kyoto cells were cultured in Dulbecco's modified Eagle's medium (DMEM; Nacalai Tesque) supplemented with 10% fetal bovine serum (FBS; Thermo Fisher Scientific, or Nichirei Biosciences Inc.). Plasmids and small interfering RNAs (siRNAs) duplexes were transfected with FuGENE-HD transfection reagent (Promega) and Lipofectamine RNAi-MAX (Thermo Fisher Scientific), respectively.

### RNA interference
The Silencer Select siRNAs targeting human ZnT4, ZnT5, ZnT6, and ZnT7 were purchased from Thermo Fisher Scientific (Supplementary Table 1). The relative amount of mRNA remaining 48 h after transfection was determined by qRT-PCR analysis using PrimePCR assays (Bio-Rad; Supplementary Table 2) and SsoAdvanced Universal SYBR Green Supermix (Bio-Rad). Thermal cycle reaction and SYBR Green detection were performed using CFX96 Touch Deep Well Real-Time PCR Detection System (Bio-Rad).

### Reagents and antibodies
4′,6-Diamidino-2-phenylindol (DAPI; Nacalai Tesque), N-ethylmaleimide (NEM; Nacalai Tesque), N,N,N′,N′-tetrakis(2-pyridylmethyl)ethylenediamine (TPEN; Dojindo Laboratories), zinc pyrithione (ZPT; Tokyo Chemical Industry), HaloTag TMR ligand (HTL-TMR; Promega), and Chelex 100 chelating resin (1432832, Bio-Rad) were commercially purchased. Antibodies were purchased as follow: mouse monoclonal antibodies against β-actin (clone AC-15; Sigma) (1:5000 for Western

blot (WB)), FLAG (HRP-conjugated, clone M2; Sigma) (1:10000 for WB), FLAG (clone M2; Sigma) (1:2000 for Immunofluorescence (IF)), GAPDH (clone GAPDH-71.1; Sigma) (1:20000 for WB), GM130 (clone 35; BD biosciences) (1:2000 for IF), golgin97 (A-21270, Invitrogen) (1:200 for IF), rat monoclonal antibody against PA (NZ-1; FUJIFILM Wako Chemicals) (1:10000 for WB and 1:5000 for IF), a rabbit monoclonal antibody against ERp44 (clone D17A6; CST) (1:1000 for WB), and rabbit polyclonal antibodies against mannosidase II (AB3712, SIGMA) (1:200 for IF), giantin (924302, BioLegend) (1:2000 for IF), GM130 (PM061; MBL) (1:2000 for IF), SEC24C (HPA040196, SIGMA) (1:500 for IF), βCOP (ab2899, Abcam) (1:1000 for IF), a DDDDK-tag (PM020, MBL) (1:4000 for IF), and ZnT6 (HPA057328; Atlas Antibodies) (1:500 for IF and 1:1000 for WB). The anti-ERp44 mouse monoclonal antibodies (clone 36C9 and 2D5) (36C9; 1:1000 for WB, 2D5; 1:50 for IF) were previously described[10,15]. The anti-ZnT7 rabbit polyclonal antibody was raised against a synthetic peptide composed of residue 2–16 of human ZnT7 (PLSIKDDEYKPPKF) (eurofins genomics) and affinity-purified. DYKDDDDK-conjugated beads (clone 1E6) (1:200 for immunoprecipitation) were purchased from FUJIFILM Wako Chemicals.

### Plasmid construction
The cDNAs coding full-length human ZnT4, ERGIC-53, or TPST2 and a part of GalT (1–61) or mannosidase II (ManII) (1–117) were amplified from human cDNA library obtained from HEK293T or HeLa Kyoto cells. The cDNAs coding ZnT5, ZnT6, or ZnT7 were codon-optimized for expression in human cell (Thermo Fisher Scientific). PCR-amplified cDNA fragments or Halo-tag were subcloned into pEF4 (Thermo Fisher Scientific) or pcDNA3.1(+)-PA plasmids using In-Fusion HD cloning kit (Takara Bio Inc.). Point mutations were introduced using a PrimeSTAR mutagenesis basal kit (Takara Bio Inc.) with appropriate sets of primers. Construction of mouse Prx4-3×FLAG was described previously[51]. Used primers in this study are summarized in Supplementary Table 3.

### Immunoblotting
To analyze disulfide-linked oligomers involving ERp44 under ZnT knockdown conditions (Fig. 5a), HeLa Kyoto cells were transfected with siRNAs and cultured for 48 h. After additional 3 h incubation in fresh media, cells were washed with PBS twice and lysed in ice-cold 10% TCA (trichloroacetic acid) as described previously[52]. Precipitants were washed with acetone three times and dissolved in SDS-NEM buffer (100 mM Tris-HCl, pH 6.8, 2% SDS, and 50 mM NEM) with sonication. Total protein concentration was determined by BCA method, and 10 μg of proteins were resolved by SDS-PAGE and analyzed by Western blot analyses. Signal intensities of immunoreactive bands were quantified using Image Lab software (Bio-Rad).

### Immunofluorescence
To analyze the effects of siRNAs on the intracellular localization of ERp44, HeLa Kyoto cells plated on coverslips were transfected with 5 nM siRNAs and incubated for 48 h. Cells were washed with PBS twice, fixed with 4% paraformaldehyde/PBS for 15 min, permeabilized by 0.1% TritonX-100/PBS for 15 min, and blocked with 2% FBS/PBS for 1 h. Coverslips were incubated with anti-ERp44 mouse monoclonal and anti-GM130 rabbit polyclonal antibodies diluted in Signal Enhancer HIKARI for Immunostain Solution A (Nacalai Tesque) overnight at 4 °C. CF488A- (20014-1), CF568- (20101-1), or CF633- (20121-1) conjugated anti-mouse IgG (Biotium), and CF488A- (20015-1) or CF568- (20103-1) conjugated anti-rabbit IgG antibodies (Biotium) were used as secondary antibodies. Nuclei were stained with DAPI. For knockdown/rescue experiments, anti-PA labeled with CF405M using Mix-n-Stain Antibody Labeling Kit (Biotium) and Alexa Fluor 594-conjugated anti-DDDDK-tag antibodies (M185-A59; MBL) were added to the cells after washing secondary antibody. Fluorescence images were acquired using a laser scanning confocal microscopy (FV1000 with FluoView software ver. 4.2, Olympus, or LSM980 with ZEN software ver.

3.4.91.00000, Carl Zeiss) equipped with UPLSAPO 60x silicon oil-immersion objective lens (NA 1.30) or C Plan-Apochromat 63×oil-immersion objective lens (NA 1.40). Pearson's correlation coefficient values were calculated using JACoP plugin on ImageJ software.

To observe fluorescence signals from the Golgi mini-stacks, HeLa Kyoto cells were plated on Carl Zeiss Cover Glasses, High Performance (474030-9000-000, Carl Zeiss) and transfected with Halo-ERGIC-53, ManII-Halo, TPST2-Halo, or GalT-Halo. After 36 h incubation, cells were washed with Chelex-treated HHBSS twice, and Halo-proteins were labeled with 5 nM HTL-TMR in serum- and phenol red-free DMEM for 30 min. Cells were then washed with Chelex-treated HHBSS twice, and subsequently treated with 33 µM nocodazole or vehicle (DMSO) for 4–6 h in complete media. Cells were washed with PBS twice, fixed with 4% PFA/PBS for 10 min (for IF of GM130 and giantin) or ice-cold methanal for 15 min at −30 °C (for IF of endogenous ManII and golgin97), and treated with antibodies diluted in Signal Enhancer HIKARI for Immunostain Solution A (for GM130 and giantin) or Signal Enhancer HIKARI for Immunostain Solution B (Nacalai Tesque, for ManII and golgin97) overnight at 4 °C. Secondary antibodies were treated as above, except for using Alexa Fluor Plus 488 conjugated anti-Rabbit IgG (A32731, Thermo Fisher Scientific). Coverslips were mounted onto slide glasses using ProLong Glass Antifade Mountant (P36982, Thermo Fisher Scientific). Fluorescence images were obtained by Airyscan super-resolution microscope system (Carl Zeiss) comprised on a Zeiss LSM880 confocal microscope or Airyscan2 super-resolution microscope system comprised on a Zeiss LSM980 confocal microscope equipped with an oil-immersion objective lens (C Plan-Apochromat 63×, NA 1.4, Oil DIC UV-VIS-IR M27). Pixel size of images was 47 nm or 42 nm, and the z-step of image stacks was 185 nm or 150 nm. Image stacks were subjected to Airyscan processing. Chromatic shifts were corrected by Chromagnon software[24]. Fluorescence images of 100 nm Tetraspeck beads (T2729, Thermo Fisher Scientific) were used as reference. The localizations of ZnT6 and ZnT7 relative to cis- and pre-cis- or medial-Golgi markers were observed and analyzed for grouping manually.

To assess the ER/ERES localization of ERp44 (Supplementary Fig. 9C), HeLa Kyoto cells transfected with Halo-STIM1(NN) were labeled with HTL-TMR and fixed with glyoxal solution as previously reported[53]. Permeabilization, blocking, and antibody treatment were performed as above. Fluorescence images were acquired with Airyscan2 super-resolution microscope comprised on LSM980 equipped with C Plan Apochromat 63x oil immersion lens (NA 1.40).

## [Zn$^{2+}$] measurement

[Zn$^{2+}$] measurement with ZnDA-1H was performed as described previously[18,20] with several modifications. HeLa Kyoto cells ($0.2 \times 10^5$) were plated onto glass-bottomed dish and transfected with 5 nM siRNAs. After 12 h incubation, 200–250 ng of plasmids for expressing Halo-tagged protein were transfected, and cells were incubated for additional 36 h. To label Halo-proteins, cells were washed with Chelex-treated HEPES-buffered HBSS[54] (HHBSS; 30 mM HEPES-NaOH, pH 7.4, 5.36 mM KCl, 137 mM NaCl) twice, and incubated with serum- and phenol red-free DMEM containing 250 nM ZnDA-1H and 5 nM HTL-TMR for 30 min. After incubation, cells were washed with Chelex-treated HHBSS twice, and culturing medium was exchanged with prewarmed Imaging buffer (Chelex-treated HHBSS supplemented with 3.0 mg/mL glucose, 1.8 mM Ca(CH$_3$COO)$_2$, and 1 µM ZnSO$_4$). Time-lapse images were acquired for 20 min at 30 s intervals by laser scanning microscopy (FV1000) equipped with UPLSAPO 60x silicon oil-immersion lens (NA 1.30). ZnDA-1H and TMR were excited by 473 nm and 559 nm laser, respectively. Cells were then fixed with 4% PFA/PBS for 10 min and washed with Chelex-treated HHBSS three times. To obtain additional images of zinc-saturating and zinc-depriving conditions, fixed cells were incubated with Chelex-treated HHBSS containing 25 µM ZPT and subsequently treated with 100 µM TPEN,

respectively. Fluorescence intensities of ZnDA-1H and TMR were quantified by ImageJ software, and [Zn$^{2+}$] was determined using the following Eq. (1):

$$\left[ Zn^{2+} \right]_{labile} = K_d \frac{R - R_{min}}{R_{max} - R}. \tag{1}$$

where $R$ is the signal ratio of ZnDA-1H/TMR, $R_{min}$ is the minimum value of $R$, $R_{max}$ is the maximum value of $R$, and $K_d$ values of ZnDA-1H are 0.35 nM for pre-cis-Golgi (pH 7.0), 0.54 nM for cis-/medial- and medial-Golgi (pH 6.5), and 0.90 nM for trans-Golgi (pH 6.0).

To measure the [Zn$^{2+}$]$_{cyto}$, HeLa Kyoto cells were transfected with 250 ng of Halo-NES (nuclear export signal). Time-lapse imaging was performed as previously reported[19]. Briefly, HaloTag proteins were labeled with 42 nM of ZnDA-2H and 5 nM of HTL-TMR for 30 min. Fluorescence images were obtained every 20 s for 25 min with Zeiss LSM980 laser scanning confocal microscope equipped with Plan Apochromat 40x lens (NA = 0.95). At 5 min and 15 min, TPEN (final 10 µM) and ZPT (final 5 µM) with ZnSO$_4$ (final 50 µM) were added to the dish, respectively. The [Zn$^{2+}$]$_{cyto}$ was calculated as same as above. A $K_d$ value of ZnDA-2H is 2.7 nM (pH7.4).

## pH measurement for organelle lumens

The pHluorin2 cDNA was amplified from the pME pHluorin2 plasmid and fused to ERGIC-53, ManII, TPST2, GalT, and STIM1(NN) by InFusion HD for the purpose of targeting to the pre-cis-, cis-/medial-, medial-, trans-Golgi and the ER, respectively. HeLa Kyoto cells plated onto glass bottomed dish were transfected with pHluorin2-tagged proteins and incubated for 36 h. Cells were washed twice with Chelex-treated HHBSS, and culturing media were exchanged with imaging buffer. The cells were incubated on humidified stage of microscopy at 37 °C for 15 min, and subsequently subjected to live-cell imaging. Fluorescence signals of pHluorin2 excited with 405 nm ($F_{405}$) and 473 nm ($F_{473}$) lasers were obtained, and the signal intensities were quantified using ImageJ software. For pHluorin2 signal calibration, pHluorin2 was fused with MBP on pMal vector, and MBP-pHluorin2 proteins were expressed in BL21 E. coli strain. Harvested bacterial cell pellets were homogenized by sonication in lysis buffer (50 mM Tris-HCl, pH 8.0, 300 mM NaCl, 1 mM EDTA) supplemented with protease inhibitor cocktail (Nacalai Tesque, 03969-21). Bacterial debris were clarified by centrifugation twice, and resultant supernatant was harvested. An aliquot of lysates was diluted in lysis buffer, and incubated with 100 µL of amylose resin (NEB, E8021) with rotation at 4 °C overnight. The beads were thoroughly washed with lysis buffer, divided into several tubes, and washed with each pH calibration buffer (10 mM bis-Tris-HCl, pH 5.56–8.31, 150 mM NaCl) twice. The beads in pH calibration buffer were transferred onto glass bottomed dish and its $F_{405}$ and $F_{473}$ signals were obtained by microscopy with identical parameters for cell image acquisition.

## Secretion assay

HeLa Kyoto cells ($1.5 \times 10^5$) plated on 60-mm plate were transfected with 5 nM siRNAs. After 6 h incubation, cells were transfected with Halo-ERp44, Ero1α (C99A, C104A)-FLAG, ERAP1-FLAG, or mPrx4-3×FLAG and incubated for additional 36 h. Cells were washed twice and incubated in Opti-MEM for 4 h. Conditioned media were collected and clarified by centrifugation. To concentrate the secreted Halo-ERp44 or mPrx4-3×FLAG proteins, aliquots of resultant supernatants were precipitated by mixing with the same volume of ice-cold 10% TCA. Precipitants were washed with acetone and dissolved in 1 × SDS-loading buffer. To concentrate FLAG-tagged ERp44-client proteins, clarified conditioned media were incubated with anti DYKDDDDK tag antibody beads (Fujifilm Wako Pure Chemicals, 018-22783) at 4 °C with rotation for 3.5 h. Beads were then washed with PBS-T three times, and precipitated proteins were eluted with 1x SDS-loading buffer. To analyze

intracellular proteins, cells remaining on plates were lysed in 1 × SDS-loading buffer and homogenized by sonication. Samples were denatured at 70 °C for 10 min, and subjected to SDS-PAGE under reducing conditions. Protein bands were visualized by immunoblotting analyses and quantified by Image Lab software.

### RUSH assay

To perform the RUSH assay, HeLa Kyoto cells ($0.4 \times 10^5$) were plated onto 15-mm round coverslips in a well of 6-well plate, and transfected with 5 nM siRNAs against the ZnTs. After 6 h incubation, 300–400 ng of pRUSH-ERp44 plasmids were transfected and incubated for another 36 h. Culturing media were exchanged to 900 μL of DMEM containing 10% streptavidin beads-pretreated FBS and cells were cultured for 3 h. Cells were then stimulated by 100 μL of DMEM containing 400 μM biotin for 0, 15, 30, 45, or 60 min, washed with PBS twice, and fixed with 4% PFA/PBS for 10 min. Fixed cells were subjected to immunofluorescence analyses as above. To determine which cell was transfected, 10 ng of LifeAct-Venus was cotransfected with pRUSH-ERp44 (ΔRDEL) as a transfection marker, and cells with similar expression level of LifeAct-Venus were randomly chosen and analyzed. To quantify the relative signal intensity of SBP-FLAG-ERp44 on the Golgi, the Golgi region was determined from fluorescence images of GM130. Signal intensities of SBP-FLAG-ERp44 on the Golgi ($F_{Golgi}$) and total cell ($F_{Total}$) were measured with Fiji, and its ratio ($F_{Golgi/Total}$) was calculated.

### Time-lapse imaging of RUSH assay

For the time-lapse imaging under the ZnTs knockdown conditions (Fig. 7 and Supplementary Fig. 15), HeLa Kyoto cells plated on 35-mm glass bottom plates were transfected with siRNAs of target ZnTs as above. Cells were further cotransfected with a combination of pRUSH-ST-SBP-EGFP and ManII-Halo or a combination of pRUSH-Halo-ERp44 and ManII-pHluorin2. ManII was used as a Golgi marker protein. After 36 h incubation, Halo proteins were labeled with 5 nM HTL-TMR for 30 min, and culturing media were exchanged to 900 μL of phenol red-free DMEM supplemented with 10% streptavidin beads-pretreated FBS, 20–50 mM HEPES-NaOH, pH 7.2, and 4 mM L-glutamine and cells were cultured for additional 3 h. The glass bottom plate was replaced onto 37 °C-prewarmed stage of microscopy and subjected to image acquisition. Fluorescence images were acquired every 1 min for 65 min with Airyscan2 multiplex 4Y mode on LSM980 (Carl Zeiss) equipped with Plan Apochromat 40x lens (NA = 0.95). At 5 min, 100 μL of phenol red-free DMEM containing 400–800 μM biotin was added to the dish (final 40–80 μM). Images were subjected to Airyscan processing. The Golgi area was determined based on the ManII-signal, and fluorescence intensities of TMR and EGFP or pHluorin2 in the Golgi area were measured using a Fiji software. Relative intensities of the ST6-SBP-EGFP and SBP-Halo-ERp44 signals on the Golgi area were calculated, and the fitting curve and $t_{1/2}$ values were obtained by Plateau followed by one phase association function on Prism software. The $v_0$ values were estimated using first 4 points of the obtained fitting curves.

### Determination of apparent $K_d$ for $Zn^{2+}$ of Halo–ZnDA-1H

The apparent $K_d$ for $Zn^{2+}$ values of Halo–ZnDA-1H at pH 5.5 and 6.0 were determined by the method described in the previous paper[18,19]. Briefly, 64 μM of purified HaloTag protein was incubated with 70 μM of ZnDA-1H in Chelex-treated HEPES buffer (100 mM, pH 7.4) for 30 min at 37 °C under dark conditions. To remove the unreacted ZnDA-1H, ultra-filtration was performed using Chelex-treated HEPES buffer (100 mM, pH 7.4). The fluorescence titration of Halo–ZnDA-1H (100 nM) was performed under the control of free $Zn^{2+}$ concentrations ranging from 15 nM to $1.1 \times 10^4$ nM at pH 5.5 and from 6.9 nM to $4.3 \times 10^3$ nM at pH 6.0 using the nitrilotriacetic acid (NTA)-$Zn^{2+}$ buffer[55] (100 mM MES, $I = 0.1$ M (NaNO₃), pH 5.5 or 6.0) at 37 °C. The apparent $K_d$ values were determined by changes in fluorescence intensity at 508 nm.

### Quantification and statistical analysis

Statistical analyses were performed with Prism software version 7.0a, 9.2.0, or 9.3.0 (GraphPad Software), using one-way ANOVA followed by Dunnett's test or Tukey's test for comparison of multiple datasets. Data represent the means of the indicated number of independent experiments. Error bars indicate the standard error of the mean (SEM) or the standard deviation (SD). $p < 0.05$ was considered to be significant.

### Reporting summary

Further information on research design is available in the Nature Portfolio Reporting Summary linked to this article.

## Data availability

The datasets generated during and/or analyzed during the current study are available in the supplementary information. Source data are provided with this paper. All other data supporting the conclusions and findings of this study are available from the corresponding author on request. Source data are provided with this paper.

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

## Acknowledgements

HeLa Kyoto cell was a gift from Kozo Tanaka (Tohoku University). LifeAct-Venus plasmid was a gift from Kazumasa Ohashi (Tohoku University). pME pHluorin2 plasmid was a gift from David Raible (Addgene plasmid # 73794;RRID: Addgene_73794). Str-STIM1-NN_ST-SBP-EGFP was a gift from Franck Perez (Addgene plasmid # 65262;RRID: Addgene_65262). We also thank to Tadayoshi Murakawa (Tokyo Institute of Technology) for plasmid construction, Misaki Kinoshita (Tohoku University) and Wakana Yamamoto (Tohoku University) for technical assistance. This work was

supported by AMED-CREST (21gm1410006h0001) to K.I. and S.M., JSPS KAKENHI (18H03978 and 21H04758 to K.I., JP19K16065, JP20H05494, and JP22H04797 to Y.A., and JP20K05702 to T.K.), and Asahi Glass Foundation (to Y.A.).

## Author contributions

Y.A. and K.I. designed the research. Y.A. performed almost all cell experiments. M.Y. performed cell experiments and plasmid constructions. T.K., T.W., R.L., Y.D., and S.M. synthesized and validated ZnDA-1H and ZnDA-2H probes. S.N. and J.K. supported super-resolution microscopy analyses. Y.A. and S.N. analyzed cell image quantification. S.W. prepared cDNAs of ZnT5, ZnT6, ZnT7 and ERGIC-53, and aided in interpreting the results. T.T. obtained preliminary data for cell experiments. Y.A. wrote the initial draft. Y.A., T.K., S.N., S.W., J.K., T.A, R.S., S.M., and K.I. wrote the manuscript. K.I. supervised the research.

## Competing interests

The authors declare no competing interests.
