## [Peer Review File · Nature Communications]

Zinc homeostasis governed by Golgi-resident ZnT family members regulates ERp44-mediated proteostasis at the ER-Golgi interfaceREVIEWER COMMENTS

Reviewer #1 (Remarks to the Author):

This manuscript quantifies the Zn²⁺ concentrations in Golgi mini-stacks using the HaloTag-targeted Zn²⁺ probe ZnDA-1H which was previously developed in the same lab.

The results are intriguing that reduction in the Golgi ZnT transporters can alter ERp44 distributions and the secretion of ERp44 client proteins. But the different roles of Zn²⁺ in subregions of Golgi in mediating ERp44 localization and functions are not entirely clarified and need further supporting data.

Here are my major comments:

1. When HaloTag is fused with the membrane proteins ERGIC-53 and TPST, the HaloTag is facing to the Golgi lumen or the cytosol? This is important to note to ensure that you are measuring the Zn²⁺ concentrations in the Golgi lumen, not the cytosol.
2. There are big variation on the values of Zn²⁺ measurements. Are Zn²⁺ concentrations among different Golgi sub-stacks significantly different from each other?
3. It is interesting that ZnT4 knockdown increases Zn²⁺ concentrations in pre-cis-Golgi, but reduces Zn²⁺ in medial and trans-Golgi? Why? Will overexpress of ZnT4 reduce Zn²⁺ in the pre-cis-Golgi?
4. It is important to compare Zn²⁺ concentration in Golgi versus cytosol. Please give measurement of Zn²⁺ in cytosol using the same ZnDA probe. Given that the K_d of ZnDA is 0.3 μ M, which is much higher than cytosol Zn²⁺ concentration, it might be difficult to take an accurate measurement of Zn²⁺. I would suggest that authors can measure Golgi sub-stacks and cytosol in cells pretreated with 100 μ M ZnCl₂ for 1 hours to load a small amount of Zn²⁺ into the cytosol and Golgi. The same measurement can also be performed in cells silenced with different ZnT transporters. These results can provide better understanding about the roles of different ZnT transporters play in regulating Golgi Zn²⁺ homeostasis.
5. In Figure 1, the effects of ZnT4 silencing are opposite of ZnT5/6 and ZnT7 silencing. What's the Zn²⁺ concentration in cells with silencing of ZnT5/6/7? Will silencing of ZnT5/6/7 be able to deplete all Golgi Zn²⁺ concentrations?
6. In Figure 2, when examining the localization of ZnT transporters in Golgi subregions with the super-resolution microscopy, please give both en face and side view of the Golgi mini-stack. Please use higher magnification figure in Supplementary Fig4 to demonstrate the Golgi cisterna.
7. The ZnT6 and ZnT7 subcellular localization are determined by the beautiful super resolution imaging results. How about the localization of ZnT4? It does not look like to be localized on Golgi from Fig. 3C.
8. It is very clear that ZnT4 and ZnT5/6 knockdown reduced the Golgi localization of ERp44, but it is not clear if ERp44 is relocated back to ER. The results from Fig.3 does not look like ER localization. Please use ER and ERGIC markers to determine the localization of ERp44 after silencing of ZnT4 and ZnT5/6.

9. It is interesting that ZnT7 and ZnT5/6 have opposite effects on ERp44 localization. What are the effects of ZnT7 and ZnT5/6 overexpression on ERp44 distribution?
10. Why does ZnT7 and ZnT5/6 affect ERp44 localization differently? Figure 4 elegantly demonstrates that increases in pre-cis Golgi Zn²⁺ concentrations reduces Golgi localization of ERp44. How about expressing ZnT5/6 in cells with silencing of all four ZnT proteins? Will it increase Golgi localization of ERp44?
11. How does HADA mutation affect ZnT7 permeability to Zn²⁺? Is ZnT7 HADA mutant partially permeable to Zn²⁺, which might explain why it can partially rescue the ERp44 location?
12. Will TPEN facilitate the Golgi translocation of ERp44?
13. How does Zn²⁺/pyrithione and TPEN affect secretion of ERp44 client protein Ero1 α , ERAP1 and Prx4?
14. Please explain whether and how the disulfide-linked complexes are related to client protein secretion.
15. For Fig 6, please use ER/ERGIC marker and endosome markers to determine if the reduced Golgi signals are due to retrograde transport of ERp44 to ER or movement to the secretory pathway. Again, rescue experiments here with expression of ZnT7 or ZnT5/6 in cells silenced with all four ZnTs might provide more evidence to elucidate the roles of different ZnT proteins in regulating ERp44 translocation.

Minor comments:

1. Please give the full name of the acronym the first time it occurs within the paper: ERp44
2. Please provide more details about how to measure the K_d of ZndA-1H. What's NTA-Zn²⁺ buffer? What is the range of Zn²⁺ concentrations used to measure K_d? Have you verified the Zn²⁺ concentration of NTA-Zn²⁺ buffer using a well-accepted Zn²⁺ sensor such as FluoZin3? The reference 19 listed at line 562 is not correctly used because reference 19 did not talk about the method about determination of K_d.

Reviewer #2 (Remarks to the Author):

The manuscript by Amagai et al reports on the role ZnTs in the Golgi and their role in regulating the function of ERp44. They show that regulation of Zn²⁺ import into the Golgi regulates the cycling of ERp44 between the ER and the Golgi and affects its ability to retrieve its clients from the Golgi to the ER.

Our understanding of the role of Zn²⁺ in the early secretory pathway is limited and therefore this work expands our understanding of this enigmatic part of cell biology. Although the data are generally of high quality, there are several inconsistencies and issues that must be addressed before I can recommend this work for publication.

1- Figure 1: I could not understand how some of the statistical differences were made. The most striking one is probably in Figure 1C (the panel with ManII-Halo). How can it be that there is a difference between Control and Znt4 depletion? I tried to extrapolate these values and calculate it myself, but there is no difference. Of note, I used a t-test and the authors used an ANOVA, which is even less likely to show a difference, given that a correction for multiple comparisons is made. Another difference is in the panel with TPST2-Halo, where I question whether there is a difference between control and ZnT7 depletion.

2- Does the depletion of one ZnT affect the localization of the other ZnTs? Some of the inconsistencies might be explained by changes in the subcellular distribution. This can be addressed by staining for ZnTs and if possible to incorporate the ZnTs into the RUSH system and investigate whether they affect the trafficking of one another.

3- I liked very much the fact that the authors used the RUSH system to investigate ERp44 trafficking. However, I think the work would benefit a lot from testing more general effects of these transporters on ER-Golgi bidirectional trafficking.

4- It remains unclear why depletion of ZnT4 increases the levels of Zn²⁺ in the ERGIC (or pre-cis-Golgi as the authors call it). Given that the ERGIC is upstream of all subsequent Golgi sub-compartments, I would have expected that the levels of Zn²⁺ are higher (or at least unchanged) in the cis- to trans-Golgi. Yet, what appears to happen is that Zn²⁺ is much lower in the medial Golgi, unchanged in the cis-Golgi and only weakly affected in the trans-Golgi. It is very hard to make sense of this.

5- In general, the effects of Zn²⁺ in the trans-Golgi are much weaker than in earlier compartments. Is there a reason for this?

6- Does Znt7 affect the function of COPI and/or KDEL-R? The authors interpret the results of Figures 3 and 6 such that ZnT7 depletion causes a mislocalization of ERp44 to the Golgi because of an effect on the retrieval of this chaperone. Is this due to a general effect on Golgi-to-ER trafficking? If the authors would stain for COPI at the Golgi, would they see less COPI recruited to this compartment?

7- I do not understand the discrepancy in Figure 5. ZnT7 depletion affects retrieval of ERp44 and oligomer formation. Consequently, ERp44 is secreted out of the cell in ZnT7 depleted cells. However, it seems to have almost no effect on the secretion of the clients of ERp44. How is this possible. On the other hand, Znt5/6 depletion results in secretion of ERp44 clients, but not of the ERp44 itself, or on

oligomer formation and on localization. These data are difficult to make sense of. Maybe I am missing something, and a simple explanation of the authors might dispel all my doubts.

8- A possible conclusion for ZnT4 is that its depletion accelerates the retrograde transport of ERp44. Looking at the data in Figure 6, this is the most likely explanation, and this is also what the authors propose. Is this a general effect, or is it specific for ERp44? This would also explain the higher Zn²⁺ levels in the ERGIC. The retrieval of ERp44 is dependent on KDEL-R and I find it hard to imagine that this effect is specific and has no effect on other clients

9- I think the conclusion the authors derive for ZnT5/6 depletion in Figure 6 is unlikely to be correct. The authors conclude that ZnT5/6 depletion has no effect on ER export of ERp44. However, I think that a traffic defect out of the ER is the best explanation for the data. When using the ERp44 mutant lacking the RDEL motif (Fig 6D), there appears to be no GFP signal appearing in the Golgi area at any time point. The authors say that this is because the rate of transport to the Golgi is the same as out of the Golgi. I think this is not likely. More likely is a defect in trafficking out of the ER. I think it is required to stain for ERES markers and perform RUSH assays with other proteins in ZnT depleted cells to make the data less ambiguous.

Overall, this is an important work, and these additional experiments would improve the overall quality and improve the clarity of the data and the conclusion derived from them.

Reviewer #3 (Remarks to the Author):

This study provides an impressive set of data that reveals how ion transport, membrane trafficking and quality control processes cooperate to support important cellular function. Using a wide range of methods, the authors show a mechanistic link between the activities of Golgi-located Zn transporters ZnT4, ZnT5/ZnT6 and ZnT7 and regulation of ERp44-mediated proteostasis. The data provided suggest that: (i) ZnT4, ZnT5/ZnT6 and ZnT7 regulate sub-organelle Zn levels in trans-, medial and cis- (pre-cis-Golgi compartments); (ii) suppression of each ZnT has a different impact on the localization and activity of ERp44; and (iii) Zn²⁺ operates as a more dominant ERp44 regulator than pH. Altogether, the main findings of the paper uncover novel Zn-dependent mechanisms that regulate proteostasis in the early secretory pathway. Therefore, the manuscript might be of great interest to the readership of Nature Communications. However, in my view, the manuscript should be revised to address the following comments.

1) This paper demonstrates that ZnT7 suppression inhibits Golgi-to-ER retrieval of ERp44 more than ZnT5/6 or ZnT4 knockdown. In light of this finding, why does ZnT7 silencing have a limited impact on ER

retention of ERp44 substrates (ERAP1 or Ero1 α), while substrate secretion increases upon ZnT4 or ZnT5/6 depletion?

2) The authors claim that Zn prevails over pH as a determinant for Erp44 retrieval from Golgi to ER. In this context, it would be nice to evaluate how silencing of each ZnT affects the interaction between ERp44 and KDELR. I would like to encourage the authors to test this using immunoprecipitation experiments (or any other protein/protein interaction approach).

3) I have a concern regarding use of Giantin as a marker of medial Golgi for analysis of intra-Golgi distribution of halo-tagged constructs or ZnTs. EM data clearly suggest that Giantin is distributed across the rims of almost all Golgi cisternae, including cis and trans. It would be safer to use Man II where possible with ZnTs or halo-tag constructs. The authors may also consider using Golgin97 for trans-Golgi compartment labelling.

4) The discussion of the cisternae maturation mechanism of ZnT distribution across the Golgi stack is confusing. Initially the authors say that ZnTs are excluded from the giantin-positive rims, where intra-Golgi trafficking of resident Golgi proteins occurs by Cop-I vesicles (according to the cisternae maturation model). Later in the discussion, they say that transporters are detectable in COP-I vesicles. These two statements contradict each other. I would avoid talking about cisterna maturation because the mechanisms of intra-Golgi trafficking still remain controversial. Instead, it would be better to discuss why some ZnTs prefer cis- or mid- or trans-Golgi. Is this defined by any specific signals, interactions or transmembrane domain length?

5) Along the same line, I noted that in addition to the Golgi, ZnT4 also localizes at numerous peripheral spots (Fig. 3C). To which compartment do these spots correspond? This information might be important to understand if ZnT4 shows preferential affinity for trans-Golgi.

6) Finally, it would probably be worth discussing the interplay between Golgi-associated ZnTs and ERp44-driven homeostasis in a broader physiological context. It is well known that loss of function of some ERp44 client proteins (like SUMF1) leads to very severe phenotypes. Some genetic disorders and/or mice phenotypes have been associated with ZnTs 4, 5 and 7 mutant and knockouts. Are there any features that are shared by ERp44 client- and ZnT-loss of function phenotypes? If not, could it be due to functional redundancy between Golgi-associated ZnTs?

Title: Zinc homeostasis governed by Golgi-resident ZnT family members regulates ERp44-mediated proteostasis at the ER-Golgi interface

Authors: Amagai Y. et al.

Manuscript No.: NCOMMS-21-48453

Reviewer #1

This manuscript quantifies the Zn²⁺ concentrations in Golgi mini-stacks using the HaloTag-targeted Zn²⁺ probe ZnDA-1H which was previously developed in the same lab. The results are intriguing that reduction in the Golgi ZnT transporters can alter ERp44 distributions and the secretion of ERp44 client proteins. But the different roles of Zn²⁺ in subregions of Golgi in mediating ERp44 localization and functions are not entirely clarified and need further supporting data.

Response: We are grateful to this reviewer for a number of constructive and useful comments for our paper. To address all comments appropriately, we have performed a lot of additional experiments and revised the manuscript with greatest care. Our point-by-point responses are described below.

Major comments:

1. When HaloTag is fused with the membrane proteins ERGIC-53 and TPST, the HaloTag is facing to the Golgi lumen or the cytosol? This is important to note to ensure that you are measuring the Zn²⁺ concentrations in the Golgi lumen, not the cytosol.

Response: HaloTag was inserted immediately after the signal sequence of ERGIC53, and into the C-terminal end of TPST2 to obtain Halo-ERGIC53 and TPST2-Halo, respectively. Based on the predicted topology models of ERGIC53 and TPST2, HaloTag is likely facing to the Golgi lumen in either construct. In support of this, our quantitative zinc imaging using these two fusion proteins revealed that [Zn²⁺] in the Golgi is 60–100 nM, which is much higher than the cytosolic [Zn²⁺] of ~500 pM, a value determined by

expressing a Halo protein with a nuclear export signal sequence and using another zinc probe ZnDA-2H with higher Zn²⁺-binding affinity than ZnDA-1H (new Supplementary Fig. 6A). Thus, it is most likely that the HaloTag domain fused with ERGIC-53 and TPST2 locates in the luminal side of the Golgi apparatus. We added this supplementary explanation in the revised manuscript (lines 103–106).

2. There are big variation on the values of Zn²⁺ measurements. Are Zn²⁺ concentrations among different Golgi sub-stacks significantly different from each other?

Response: To address this comment, we made statistical comparison of the labile Zn²⁺ concentrations between the different Golgi cisternae in siControl cells (new Supplementary Fig. 4). While the measured [Zn²⁺] value of each Golgi cisterna was variable per cell, the statistical analysis indicated that the medial-Golgi contained significantly lower labile [Zn²⁺] than the other Golgi stacks. In contrast, the labile [Zn²⁺] in the pre-*cis*-Golgi tends to be higher than those in the other Golgi cisternae although the differences between the pre-*cis*- and *cis*-Golgi and between the pre-*cis*- and *trans*-Golgi are not statistically substantial. Based on the results of this statistical analysis, we carefully rephrased the corresponding part of the text (lines 156–160).

3. It is interesting that ZnT4 knockdown increases Zn²⁺ concentrations in pre-cis-Golgi, but reduces Zn²⁺ in medial and trans-Golgi? Why? Will overexpress of ZnT4 reduce Zn²⁺ in the pre-cis-Golgi?

Response: While this result is interesting, it is not straightforward to explain how ZnT4 knockdown increases [Zn²⁺] in pre-*cis*-Golgi while decreasing [Zn²⁺] in medial and trans-Golgi. To tackle this problem, we quantified the cytosolic [Zn²⁺] under various ZnTs-knockdown conditions, as requested by this reviewer (please see also our response to the next comment). We thus found that ZnT4 knockdown significantly increased the cytosolic [Zn²⁺] whereas the other ZnT knockdowns rather decreased it (new Supplementary Fig.

6A). In this context, a previous study reported that aberrantly increased cytosolic Zn^{2+} can be incorporated into the ERGIC vesicles (Miki et al., *Nat. Methods*, 2016). Given that, it seems possible that the $[Zn^{2+}]$ increases exclusively in the Halo-ERGIC53-localizing pre-*cis*-Golgi through the elevation of cytosolic $[Zn^{2+}]$ in ZnT4-knockdown cells.

The decrease of $[Zn^{2+}]$ in the medial-/*trans*-Golgi of ZnT4-knockdown cells suggests the direct or indirect contribution of ZnT4 to zinc homeostasis in these Golgi cisternae. Indeed, our immunofluorescence experiments revealed that while the major portion of exogenously expressed ZnT4-FLAG localizes at the recycling endosomes, early endosomes, and lysosome, some localizes around the Golgi and plasma membrane (new Supplementary Fig. 12). Thus, ZnT4 likely serves to upregulate $[Zn^{2+}]$ in the medial-/*trans*-Golgi via vesicle transports between the distal Golgi and post-Golgi compartments, and/or via direct Zn^{2+} uptake from the cytosol to the Golgi lumen.

Regarding the Zn^{2+} quantification in ZnT4-overexpressing cells, we have to reluctantly say that this experiment is practically difficult, because ZnT4-FLAG-expressing cells cannot be exclusively selected in our live-cell imaging systems. In other words, the transfection efficiency of ZnT4-FLAG was not high enough to allow the random pickup of ZnT4-expressing cells. Therefore, instead of directly measuring the $[Zn^{2+}]$ in the ZnT4-expressing living cells, we examined by immunofluorescence of the fixed cells if overexpression of ZnT4 affects the localization of ERp44. As shown in new Supplementary Fig. 13, overexpression of ZnT4 did not alter the Golgi-localization of ERp44 in almost all cells, suggesting that unlike overexpressed ZnT5/6 and ZnT7, overexpressed ZnT4 does not change the $[Zn^{2+}]$ in the pre-*cis*-Golgi. This result has been additionally described in the revised text (lines 263–268).

In summary, ZnT4 works primarily in the distal and post-Golgi for the maintenance of zinc homeostasis in the secretory pathway, and the increased labile $[Zn^{2+}]$ in the pre-*cis*-Golgi of ZnT4-silencing cells is likely a secondary effect of the elevation of the cytosolic Zn^{2+} . This issue has been additionally discussed in the first section of Discussion part (lines 455–462).

4. It is important to compare Zn^{2+} concentration in Golgi versus cytosol. Please give measurement of Zn^{2+} in cytosol using the same ZnDA probe. Given that the Kd of

ZnDA is 0.3 mM, which is much higher than cytosol Zn²⁺ concentration, it might be difficult to take an accurate measurement of Zn²⁺. I would suggest that authors can measure Golgi sub-stacks and cytosol in cells pretreated with 100 mM ZnCl₂ for 1 hours to load a small amount of Zn²⁺ into the cytosol and Golgi. The same measurement can also be performed in cells silenced with different ZnT transporters. These results can provide better understanding about the roles of different ZnT transporters play in regulating Golgi Zn²⁺ homeostasis.

Response: We appreciate this essential comment. As mentioned above, we recently established a method for quantifying the cytosolic [Zn²⁺] using newly developed ZnDA probes, ZnDA-2H and ZnDA-3H, with even higher Zn²⁺-binding affinity than the original ZnDA-1H probe (Liu et al., *ACS sensors*, 2022). We thereby investigated the effects of ZnT-knockdowns on the cytosolic Zn²⁺ concentration, as shown in new Supplementary Fig. 6A. Consequently, it has been found that ZnT4-knockdown significantly elevated the cytosolic [Zn²⁺] whereas the knockdowns of ZnT5+6, ZnT7, or ZnT4+5+6+7 rather reduced it. To further explore the physiological effects of ZnTs-knockdown, we analyzed the mRNA levels of typical zinc homeostasis genes, MT2A and ZnT1, and ZIP14 by qRT-PCR. The analyses revealed that MT2A was significantly upregulated in ZnT4-knockdown cells (new Supplementary Fig. 6B), consistent with the elevation of the cytosolic [Zn²⁺] under this condition. In contrast, the knockdown of ZnT5/6 or ZnT7 hardly altered the expression levels of any zinc homeostasis genes, suggesting that the decrease of cytosolic [Zn²⁺] in these knockdown cells was not substantial to cells. These results were shown in new Supplementary Fig. 6, and described in the revised manuscript (lines 173–188).

5. In Figure 1, the effects of ZnT4 silencing are opposite of ZnT5/6 and ZnT7 silencing. What's the Zn²⁺ concentration in cells with silencing of ZnT5/6/7? Will silencing of ZnT5/6/7 be able to deplete all Golgi Zn²⁺ concentrations?

Response: To answer this question, we measured the labile [Zn²⁺] at each Golgi cisterna under the ZnT5/6/7 triple-knockdown condition (new Supplementary Fig. 5). Consequently, we found that ZnT5/6/7-knockdown significantly decreased the Zn²⁺

concentration in all Golgi cisternae as did the knockdown of all of ZnT4/5/6/7. Thus, ZnT5/6/7 seems to predominantly govern zinc homeostasis throughout the Golgi apparatus, although the ZnT4-mediated zinc regulation cannot also be ignored. This result has been described in the revised manuscript (lines 163–165).

6. In Figure 2, when examining the localization of ZnT transporters in Golgi subregions with the super-resolution microscopy, please give both en face and side view of the Golgi mini-stack. Please use higher magnification figure in Supplementary Fig4 to demonstrate the Golgi cisterna.

Response: To demonstrate the Golgi mini-stacks viewed from multiple angles, we reconstructed 3D projection images (new Supplementary Movies 1–4). The 3D projection images clearly visualize the relative localizations of ZnTs in the Golgi cisternae and reinforce the *cis*-/medial- and pre-*cis*-/*cis*-Golgi localizations of ZnT6 and ZnT7, respectively.

Considering that the Golgi cisternal maturation model is still controversial, we removed the original Supplementary Fig. 4E and 4F and the discussion regarding the mechanism of ZnTs localization based on that model in the revised manuscript. This revision is also a request by Reviewer #3 (please see also Comment 4 of Reviewer #3).

7. The ZnT6 and ZnT7 subcellular localization are determined by the beautiful super resolution imaging results. How about the localization of ZnT4? It does not look like to be localized on Golgi from Fig. 3C.

Response: To address this comment, we attempted to immunostain endogenous ZnT4 using two commercially available anti-ZnT4 antibodies (R12-3524; Assay Biotechnology, and PA2191; Boster immunoleader) and our original anti-ZnT4 antibodies raised against three sorts of peptides corresponding to the N-terminal (residue 54–66), cytosolic loop (residue 241–257), and C-terminal (residue 381–395) segments of human ZnT4. However, none of them worked well, resulting in the failure to determine the precise localization of endogenous ZnT4. Thus, we decided to co-stain exogenously expressed ZnT4-FLAG with several organelle markers. As shown in new Supplementary Figure 12A, our

immunofluorescence experiments revealed that the ZnT4-FLAG signals did not co-localize well with either of the ER, ERGIC, COPI, or autophagosomes, but showed much better co-localization with recycling endosomes, early endosomes, and lysosomes. We also note that ZnT4-FLAG partially co-localized with *cis*-Golgi and TGN. These data suggest that exogenously expressed ZnT4 preferentially localizes in the distal-side Golgi and post-Golgi compartments, as described in the revised text (lines 255–261). We also observed certain signals of ZnT4-FLAG on the plasma membrane (new Supplementary Figure 12B). Considering that ZnT4-knockdown elevated the cytosolic Zn²⁺ concentration (Supplementary figure S6), ZnT4 likely serves to export Zn²⁺ from the cytosol to the extracellular space, as briefly discussed in the revised manuscript (lines 455–460).

8. It is very clear that ZnT4 and ZnT5/6 knockdown reduced the Golgi localization of ERp44, but it is not clear if ERp44 is relocated back to ER. The results from Fig.3 does not look like ER localization. Please use ER and ERGIC markers to determine the localization of ERp44 after silencing of ZnT4 and ZnT5/6.

Response: As requested, ZnT4- and ZnT5/6-knockdown cells were immunostained for ERp44 and ERGIC-53 to assess the co-localization of these two. As demonstrated in new Supplementary Fig. 9A & B, quantitative analyses revealed that co-localization of ERp44 with ERGIC-53 was reduced upon either ZnT4 or ZnT5/6 knockdown. To investigate if ERp44 is relocated back to the ER and ER exit site (ERES) under these knockdown conditions, cells were also stained for ERp44, exogenously expressed Halo-STIM1(NN), an ER membrane marker, and SEC24C, an ERES marker. We thus found that most ERp44 signals were associated with the ER membrane and/or ERES in ZnT4- and ZnT5/6-silencing cells (Supplementary Fig. 9C). Taken together, ERp44 seems very likely to be relocated back to the ER and/or ERES after ZnT4- or ZnT5/6-knockdown. This observation has additionally been described in the revised text (lines 224–232).

9. It is interesting that ZnT7 and ZnT5/6 have opposite effects on ERp44 localization. What are the effects of ZnT7 and ZnT5/6 overexpression on ERp44 distribution?

Response: To answer this question, we investigated the intracellular localization of ERp44 in ZnT7 or ZnT5/6 overexpression cells. As shown in Supplementary Fig. 13, overexpression of either PA-ZnT5+ZnT6-FLAG or PA-ZnT7 reduced the co-localization of ERp44 with the Golgi, suggesting that both ZnT5/6 and ZnT7 have the potential to regulate the localization of ERp44 likely through the uptake of Zn²⁺ into the Golgi lumen. This observation has been additionally described in the revised manuscript (lines 264–267).

10. Why does ZnT7 and ZnT5/6 affect ERp44 localization differently? Figure 4 elegantly demonstrates that increases in pre-cis Golgi Zn²⁺ concentrations reduces Golgi localization of ERp44. How about expressing ZnT5/6 in cells with silencing of all four ZnT proteins? Will it increase Golgi localization of ERp44?

Response: As shown in new Fig. 3c and d, the overexpression of ZnT5+6 in ZnT4/5/6/7-silencing cells reduced the Golgi localization of ERp44 as did that of ZnT7. This data clearly indicates that the ZnT5/6 complex has a potential to regulate the localization of ERp44. However, endogenous ZnT7 and ZnT5/6 are localized primarily in the pre-*cis*-/*cis*-Golgi and *cis*-/medial-Golgi, respectively, as demonstrated in Fig. 2. The depletion of Zn²⁺ in the pre-*cis*-Golgi by ZnT7 knockdown likely allows ERp44 to move to the more distal Golgi areas, whereas ZnT5/6 knockdown does not largely affect [Zn²⁺] in the pre-*cis*-Golgi (Fig. 1d) and therefore still enables ERp44 to return to the ER efficiently. These observations well reflect the different effects of the ZnT7 and ZnT5/6 knockdowns on the ERp44 localization (please see also our response to comment 7 of Reviewer #2).

11. How does HADA mutation affect ZnT7 permeability to Zn²⁺? Is ZnT7 HADA mutant partially permeable to Zn²⁺, which might explain why it can partially rescue the ERp44 location?

Response: To address this comment, we constructed a new ZnT7 mutant, ZnT7(AAAA) in which all His and Asp residues at the Zn²⁺-binding motif were mutated to Ala, and assessed its rescue ability for ERp44 relocation to the ER. As expected, ZnT7(AAAA) was unable to rescue the normal distribution of ERp44 most likely due to the lack of Zn²⁺-

transport activity (new Fig. 4a–c). We also repeated the same assay for ZnT7(ADHA) (in the revised manuscript, HADA has been renamed to ADHA according to the order of these four residues from the N-terminus). Consequently, we observed that like ZnT7(AAAA), ZnT7(ADHA) could not rescue the intracellular localization of ERp44. Based on these new data, we have updated Fig. 4b and c, and revised the corresponding part of the text (lines 272–282).

12. Will TPEN facilitate the Golgi translocation of ERp44?

Response: Yes. In our previous paper (Watanabe, Amagai et al., *Nat. Commun.*, 2019), we demonstrated that TPEN treatment dramatically relocates ERp44 to the Golgi apparatus like all ZnT4/5/6/7 knockdown.

13. How does Zn²⁺/pyrithione and TPEN affect secretion of ERp44 client protein Ero1 α , ERAP1 and Prx4?

Response: We previously reported that TPEN treatment enhanced the secretion of ERp44 client proteins, Ero1 α and ERAP1 (Watanabe et al., *Nat. Commun.*, 2019) and added this information in the revised text (lines 310–311). In the present work, we also tried to investigate the effect of Zn²⁺/pyrithione (ZPT) on the secretion of these client proteins. However, the addition of 5 μ M or 10 μ M of ZPT severely damaged the cells as shown below, not allowing us to show this requested data, unfortunately.

14. Please explain whether and how the disulfide-linked complexes are related to client protein secretion.

Response: Our previous studies indicated that ERp44, in most cases, forms disulfide-linked complexes with client proteins via its Cys-29 to retrieve them from the Golgi to the ER (Anelle et al., *EMBO J.*, 2003). More recently, our biochemical analysis demonstrated that the complex formation is stabilized by Zn^{2+} through the release of its C-terminal tail (Watanabe et al., *Nat. Commun.*, 2019). In line with this, TPEN treatment and all ZnT4/5/6/7 knockdowns greatly decreased disulfide-linked complexes in cells (Fig. 5a and b), and the latter enhanced the secretions of ERp44 itself and its client proteins (Fig. 5c-f). However, it seems that the accumulation levels of the disulfide-linked complexes do not simply correlate with the client retention activity of ERp44; for instance, whereas the knockdown of ZnT5/6 generated greater amount of the disulfide-linked oligomers than that of ZnT7 (Fig. 5a, b), the former allowed more client secretion than

the latter. One likely explanation is that Zn^{2+} in the distal Golgi area (i.e. medial-/trans-Golgi) serves as a gatekeeper for the ERp44-mediated client retention. In support of this, the knockdown of either ZnT4, ZnT5+6, or ZnT4+5+6+7 caused the marked decrease of $[Zn^{2+}]$ in this area (Fig. 1d) and led to the enhanced client secretion (Fig. 5e-j), although some disulfide-linked oligomers were still formed presumably in the pre-*cis*- and *cis*-Golgi. Conversely, the knockdown of ZnT7 compromises the formation of disulfide-linked oligomers due to the lack of Zn^{2+} in the pre-*cis*-Golgi (Fig. 5a, b), but still ensures the efficient client retention probably using Zn^{2+} in the distal Golgi area. Once the ERp44-client complex reaches the ER, the complex will be resolved immediately due to the extremely low $[Zn^{2+}]$ in the ER (Liu et al., *ACS Sensors*, 2022). To avoid readers' confusion, we have described our interpretations about this important issue in Discussion section of the revised manuscript (lines 487–506). Please see also our response to comment 7 by Reviewer #2.

15. For Fig 6, please use ER/ERGIC marker and endosome markers to determine if the reduced Golgi signals are due to retrograde transport of ERp44 to ER or movement to the secretory pathway. Again, rescue experiments here with expression of ZnT7 or ZnT5/6 in cells silenced with all four ZnTs might provide more evidence to elucidate the roles of different ZnT proteins in regulating ERp44 translocation.

Response: To clarify whether the reduced Golgi signals of ERp44 are attributed to its retrograde transport to the ER, we performed the live-cell imaging combined with the RUSH system using SBP-Halo-ERp44(WT) and its counterpart mutant lacking the ER-retrieval signal, SBP-Halo-ERp44 (Δ RDEL). Consequently, we succeeded in visualizing the biotin-triggered relocation of both the ERp44 constructs from the ER to the Golgi (new Supplementary Fig. 15 and Supplementary Movies 5 and 6). Notably, after reaching the Golgi, SBP-Halo-ERp44(WT) was seemingly kept localized around the Golgi and its peripheral wide area with network structures, whereas SBP-Halo-ERp44(Δ RDEL) displayed a strikingly different localization pattern, and its intracellular signals largely decreased during 40–60 min after the biotin stimulation (new Supplementary Fig. 15 and Supplementary Movies 5 and 6). Thus, our live-cell imaging analysis corroborated that ERp44(WT) was cycling between the ER and Golgi after reaching the Golgi, whereas

ERp44(Δ RDEL) was secreted out of the cell. These additional data have been described in the revised text (lines 342–354).

Regarding the requested rescue experiment, the simultaneous transfection of pZnTs and pRUSH plasmids was very low efficiency, resulting in the failure to perform the RUSH assay under the rescue conditions. Nonetheless, our rescue experiments using fixed cells (new Figs. 3 and 4, and Supplementary Fig. 14) and live-cell imaging (new Fig. 7) provided further evidence for the different roles of the Golgi-resident ZnTs in regulation of the ERp44 traffic in the ESP.

Minor comments:

1. Please give the full name of the acronym the first time it occurs within the paper: ERp44

Response: We added the full name of ERp44 in the revised manuscript (line 65). We also added the full names of following proteins: Ero1 α (line 68), ERAP1 (line 69), Prx4 (line 69), ERGIC53 (line 101), TPST2 (line 101), and GalT (line 102).

2. Please provide more details about how to measure the K_d of ZnDA-1H. What's NTA-Zn²⁺ buffer? What is the range of Zn²⁺ concentrations used to measure K_d? Have you verify the Zn²⁺ concentration of NTA-Zn²⁺ buffer using a well-accepted Zn²⁺ sensor such as FluoZin3? The reference 19 listed at line 562 is not correctly used because reference 19 did not talk about the method about determination of K_d.

Response: The K_d values of ZnDA-1H at pH 5.5–7.4 and Halo-ZnDA-1H at pH 6.5–7.4 were determined by our preceding work (Ref 18: Kowada, T. *et al.*, *Cell Chem. Biol.*, 2020, 27, 1521–1531.). Briefly, for this determination, we used a set of zinc ion buffers composed of different molar ratios of ZnSO₄ and nitrilotriacetic acid (NTA), a metal buffering component (Perrin, D.D. and Dempsey, B. 1974 (added as Ref. 55)). Using this NTA-Zn²⁺ buffer system, the free zinc concentrations were adjusted ranging from 15 nM to 1.1×10^4 nM at pH 5.5 and from 6.9 nM to 4.3×10^3 nM at pH 6.0. To obtain the K_d values of ZnDA-1H at pH 5.5 and pH 6.5, its fluorescence intensity was measured at various Zn²⁺ concentrations, as demonstrated below.

Change in fluorescence intensity of Halo-ZnDA-1H at 508 nm at pH 5.5

Change in fluorescence intensity of Halo-ZnDA-1H at 508 nm at pH 6.0

Using the updated protocol, we corrected the K_d value of ZnDA-1H at pH 5.5 from $1.51 \pm 0.03 \mu\text{M}$ to $1.6 \pm 0.1 \mu\text{M}$ and that at pH 6.0 from $0.86 \pm 0.09 \mu\text{M}$ to $0.90 \pm 0.07 \mu\text{M}$. As pointed out by this reviewer, our protocol for preparation of the zinc buffer was not described in Reference 19 of the original manuscript (Kowada T. *et al. STAR Protoc.* **2021**, 2, 100395), but in our more recent papers (Ref 18 and Liu, R. *et al. ACS Sens.* **2022**, 7, 748–757 added as Ref. 19). We have correctly cited the latter two papers in the revised manuscript.

As requested, we also confirmed that, using the NTA- Zn^{2+} buffer, the K_d value of FluoZin-3 at pH 7.4 was determined to be $10.5 \pm 0.2 \text{ nM}$ (Mean \pm SD, $n = 3$) as shown below, which is almost comparable to the reported value of 15 nM (Gee, K. R. *et al. J. Am. Chem. Soc.* **2002**, 124, 776).

Change in fluorescence intensity of FluoZin-3 at pH 7.4

In conclusion, the results shown above verify that our NTA-Zn²⁺ buffer provides an appropriate range of labile Zn²⁺ concentrations to determine the K_d value of the ZnDA-1H probe. We added the above information in Methods section of the revised manuscript, as below (lines 738–747):

Determination of apparent K_d for Zn²⁺ of Halo-ZnDA-1H

The apparent K_d for Zn²⁺ values of Halo-ZnDA-1H at pH 5.5 and 6.0 were determined by the method described in the previous papers^{19, 20}. Briefly, 64 μ M of purified HaloTag protein was incubated with 70 μ M of ZnDA-1H in Chelex-treated HEPES buffer (100 mM, pH 7.4) for 30 min at 37 °C under dark conditions. To remove the unreacted ZnDA-1H, ultra-filtration was performed using Chelex-treated HEPES buffer (100 mM, pH 7.4). The fluorescence titration of Halo-ZnDA-1H (100 nM) was performed using the nitrilotriacetic acid (NTA)-Zn²⁺ buffer (100 mM MES, I = 0.1 M (NaNO₃), pH 5.5 or 6.0) at 37 °C to control the free zinc concentration ranging from 15 nM to 1.1×10^4 nM at pH 5.5 and from 6.9 nM to 4.3×10^3 nM at pH 6.0. The apparent K_d values were determined by changes in fluorescence intensity at 508 nm.

Reviewer #2 (Remarks to the Author):

The manuscript by Amagai et al reports on the role ZnTs in the Golgi and their role in regulating the function of ERp44. They show that regulation of Zn²⁺ import into the Golgi regulates the cycling of ERp44 between the ER and the Golgi and affects its ability to retrieve its clients from the Golgi to the ER.

Our understanding of the role of Zn²⁺ in the early secretory pathway is limited and therefore this work expands our understanding of this enigmatic part of cell biology. Although the data are generally of high quality, there are several inconsistencies and issues that must be addressed before I can recommend this work for publication.

Response: We appreciate these overall positive comments.

1- Figure 1: I could not understand how some of the statistical differences were made. The most striking one is probably in Figure 1C (the panel with ManII-Halo). How can it be that there is a difference between Control and Znt4 depletion? I tried to extrapolate these values and calculate it myself, but there is no difference. Of note, I used a t-test and the authors used an ANOVA, which is even less likely to show a difference, given that a correction for multiple comparisons is made. Another difference is in the panel with TPST2-Halo, where I question whether there is a difference between control and ZnT7 depletion.

Response: We are grateful to this reviewer for careful review of our paper. The statistical analyses were performed using all data points obtained from randomly chosen cells. An original dataset indicated by yellow squares showed an aberrantly high Zn^{2+} concentration in the *cis*-/medial-Golgi of ZnT4-KD cells, likely causing the significant difference between siControl and siZnT4. To verify whether this difference is really significant, we repeated another set of the experiment and performed statistical analysis with it again. To address the second concern of this reviewer, we also performed another set of the experiment using TPST2-Halo. By appending these data and performing statistical analysis with one-way ANOVA followed by Tukey's multiple comparison test, we verified that ZnT4- and ZnT7-silencings did not significantly change the labile [Zn^{2+}] in the *cis*-/medial- and medial-Golgi, respectively. With these new data, we have prepared new Fig. 1d as a replacement of original Fig. 1c. Also, the text has been modified in the revised manuscript (lines 161–172).

2- Does the depletion of one ZnT affect the localization of the other ZnTs? Some of the inconsistencies might be explained by changes in the subcellular distribution. This can be addressed by staining for ZnTs and if possible to incorporate the ZnTs into the RUSH system and investigate whether they affect the trafficking of one another.

Response: To explore this interesting possibility, we investigated the effects of ZnT4- or ZnT7-silencing on the sub-organelle localization of ZnT6. As demonstrated in new Supplementary Fig. 8A-D, ZnT6 was kept localized in the *cis*-/medial-Golgi after ZnT4-

or ZnT7-knockdown. Similarly, we verified that neither ZnT4- nor ZnT5+6-knockdown altered the localization of ZnT7 (Supplementary Fig. 8E, F). These additional data have been described in the revised manuscript (lines 208–210).

3- I liked very much the fact that the authors used the RUSH system to investigate ERp44 trafficking. However, I think the work would benefit a lot from testing more general effects of these transporters on ER-Golgi bidirectional trafficking.

Response: We greatly appreciate this essential comment. To test the general effects of ZnTs-knockdown on the ER-to-Golgi anterograde trafficking, we employed another model cargo protein in which the TM domain of ST6 β -galactoside sialyltransferase (ST), one of the Golgi-resident membranous protein, was fused to the N-terminus of SBP-EGFP, and applied it for the RUSH experiments under the ZnTs knockdown conditions. We also performed the time-lapse imaging measurements to visualize the ER-to-Golgi transport of the above client in real time. As demonstrated in new Figure 7a–d, we found that half-time ($t_{1/2}$) required for the relocation of ST-SBP-EGFP to the Golgi was hardly altered by ZnTs-knockdown. This result suggests that the ER-to-Golgi anterograde trafficking per se is not affected by the $[Zn^{2+}]$ in the Golgi.

Similarly, we performed the time-lapse imaging of SBP-Halo-ERp44 in ZnTs-silencing cells. Interestingly, we found that ZnT7- or ZnT4/5/6/7-knockdown significantly extended the $t_{1/2}$ value, whereas ZnT5/6-knockdown did not show such effects. In either knockdown condition, however, the rate of the initial signal increase derived from Golgi-reached SBP-Halo-ERp44 (k_0) was not altered (Fig. 7e-i). These observations suggest that Zn^{2+} in the pre-*cis*- and *cis*-/medial Golgi do not affect the ER exit of ERp44, but controls the trafficking of ERp44 after reaching the Golgi. More concretely, ZnT7- or ZnT4/5/6/7-knockdown likely facilitates the secretion of ERp44 from the Golgi to the downstream compartments due to the lower $[Zn^{2+}]$ in the Golgi, as supported by the enhanced secretion of ERp44 under this condition (Fig. 5c). In contrast, the knockdown of ZnT4 hardly allowed the relocation of SBP-Halo-ERp44 to the Golgi (new Fig. 7f). Since $[Zn^{2+}]$ in the pre-*cis*-Golgi was significantly elevated upon ZnT4-knockdown (Fig. 1d), the retrograde transport of ERp44 from the pre-*cis*-Golgi to the ER would be greatly enhanced under this knockdown condition.

Thus, our RUSH assays demonstrated that ZnTs-knockdown did not alter the ER-to-Golgi transport of a general cargo, but the trafficking of ERp44 after reaching the pre-*cis*- or *cis*-/medial- Golgi was affected by the knockdowns of the Golgi-resident ZnTs. In other words, ERp44 is a special cargo protein whose bidirectional trafficking between the ER and Golgi and secretion out of the Golgi are regulated by Zn²⁺ in the Golgi. These findings have been described in a new Result section “Knockdowns of Golgi-resident ZnTs do not impair the ER-to-Golgi trafficking” (lines 394–428).

4- It remains unclear why depletion of ZnT4 increases the levels of Zn²⁺ in the ERGIC (or pre-cis-Golgi as the authors call it). Given that the ERGIC is upstream of all subsequent Golgi sub-compartments, I would have expected that the levels of Zn²⁺ are higher (or at least unchanged) in the cis- to trans-Golgi. Yet, what appears to happen is that Zn²⁺ is much lower in the medial Golgi, unchanged in the cis-Golgi and only weakly affected in the trans-Golgi. It is very hard to make sense of this.

Response: It is really intriguing that ZnT4-silencing elevated labile [Zn²⁺] in the pre-*cis*-Golgi. To address this enigmatic problem, we also investigated the effect of ZnTs-silencing on the cytosolic labile [Zn²⁺]. Unexpectedly, we found that ZnT4-silencing significantly increased the cytosolic [Zn²⁺]. Furthermore, our immunofluorescence study revealed that some portion of ZnT4-FLAG localizes at the plasma membrane, presumably conducting Zn²⁺ efflux from the cytosol to the extracellular space. A recent study reported that the ERGIC53-positive vesicles primarily incorporate Zn²⁺ into its lumen, when the cytosolic labile Zn²⁺ is aberrantly increased (Miki et al., 2016, *Nat. Methods*). Thus, it seems possible that the increased cytosolic Zn²⁺ caused by ZnT4-knockdown is transported specifically to the pre-*cis*-Golgi.

While metalation of secretory zinc-enzymes likely takes place in the Golgi, a previous work demonstrated that the overexpression of ZnT4 inhibits the activity of the *trans*-Golgi-resident galactosyltransferase probably because excess Zn²⁺ decreases the K_m and V_{max} for both glucose and UDP-galactose in the lactose synthase reaction (N.H. McCormick and S.L. Kelleher, 2012, *Am J Physiol Cell Physiol*). This result suggests that ZnT4 also works for Zn²⁺ uptake into the *trans*-Golgi. Consistently, our immunofluorescence study demonstrated that some portion of ZnT4-FLAG localizes in

the distal-side Golgi whereas much localizes in the post-Golgi vesicles (new Supplementary Figure 12A). Given that ZnT4 transports Zn²⁺ to the medial-/*trans*-Golgi, it seems to make sense that ZnT4-silencing reduces the [Zn²⁺] in these Golgi cisternae. Additionally, we need to note that the Golgi-resident ZIP transporters may contribute to reducing the labile [Zn²⁺] in the *cis*-/medial-/*trans*-Golgi. Future studies will provide deeper and more comprehensive insight into mechanisms of Zn²⁺ homeostasis in the ESP ensured by the cooperation of ZnTs and ZIPs. The above discussion has been added to the Discussion section of the revised manuscript (lines 455–462).

5- In general, the effects of Zn²⁺ in the trans-Golgi are much weaker than in earlier compartments. Is there a reason for this?

Response: This is an interesting issue to be discussed. Indeed, although the knockdown of all ZnT4/5/6/7 significantly decreased the labile [Zn²⁺] in the *trans*-Golgi, the extent of the decrease was smaller than those in the *cis*- and medial-Golgi. One of the possible reasons is the involvement of zinc transporters other than ZnT4/5/6/7 in zinc homeostasis in the Golgi. Consistently, ZnT10 was suggested to transport Zn²⁺ to the *trans*-Golgi in a neuroblastoma cell line (H J Bosomworth et al., 2012, *Metallomics*). ZnT4/5/6/7-silencing cells may tend to maintain Zn²⁺ homeostasis in the *trans*-Golgi via the induction of ZnT10, while the expression level of ZnT10 is very low under steady state in a HeLa cell line (The human protein atlas: <https://www.proteinatlas.org/ENSG00000196660-SLC30A10/subcellular>). Another possible mechanism is that the endosome-to-Golgi trafficking route via the retromer complex may contribute to Zn²⁺ homeostasis in the *trans*-Golgi. Although there is no evidence for this hypothesis so far, organelle-organelle communications and gene regulations for the maintenance of Zn²⁺ homeostasis in the ZnT4/5/6/7-knockdown cells are interesting subjects to be elucidated by future studies.

6- Does Znt7 affect the function of COPI and/or KDEL-R? The authors interpret the results of Figures 3 and 6 such that ZnT7 depletion causes a mislocalization of ERp44 to the Golgi because of an effect on the retrieval of this chaperone. Is this due to a general effect on Golgi-to-ER trafficking? If the authors would stain for COPI at the Golgi, would they see less COPI recruited to this compartment?

Response: To address this comment, the ZnTs-silencing cells were immunostained for β -COP, one of the COPI coatomer proteins, and GM130. As demonstrated in Supplementary Fig. 10, quantitative imaging analyses revealed that the signal intensity of β -COP on the Golgi apparatus was elevated upon ZnT4-, ZnT5/6-, ZnT7-, or ZnT4/5/6/7-knockdown. Presumably, two alternative explanations are possible for this observation; activated recruitment of the COPI coatomer proteins on the Golgi membrane, or prevented exit of the COPI-coated vesicle from the Golgi under the ZnTs knockdown conditions. In either case, the results suggest that ZnT-knockdown does not inhibit the recruitment of COPI coatomer proteins. The results have been additionally discussed in the revised text (lines 232–239).

7- I do not understand the discrepancy in Figure 5. ZnT7 depletion affects retrieval of ERp44 and oligomer formation. Consequently, ERp44 is secreted out of the cell in ZnT7 depleted cells. However, it seems to have almost no effect on the secretion of the clients of ERp44. How is this possible. On the other hand, Znt5/6 depletion results in secretion of ERp44 clients, but not of the ERp44 itself, or on oligomer formation and on localization. These data are difficult to make sense of. Maybe I am missing something, and a simple explanation of the authors might dispel all my doubts.

Response: We appreciate this critical comment. As ZnT7-silencing accumulated ERp44 primarily in the Golgi, some of ERp44 molecules could be passively leaked into the downstream secretory vesicles and secreted out of cells. Indeed, the secretion of overexpressed Halo-ERp44 was enhanced by ZnT7-silencing. However, substantial amount of exogenous and endogenous ERp44 remained intracellular (primarily in the Golgi, as revealed by our cell imaging analysis) after ZnT7-knockdown, as shown in Fig. 5c, 5e, and 5g. Moreover, since the labile $[Zn^{2+}]$ in the *cis*- to *trans*-Golgi is almost unchanged by ZnT7 knockdown, Zn^{2+} -mediated activation of ERp44 is supposed to be still possible. Given that, ERp44 could readily form disulfide-linked complexes with client proteins in the *cis*- to *trans*-Golgi, and then the complexes could be transported back to the ER via KDELR immediately. At this stage, the weakly acidic pH in the Golgi

will facilitate binding of ERp44 to KDEL. As a result, the disulfide-linked oligomers involving ERp44 are likely short-lived and hence accumulate in smaller amount in ZnT7-knockdown cells. This situation is largely different from normal or siControl cells, in which ERp44-client complexes are formed primarily in the pre-*cis*- and *cis*-Golgi and may likely be transported back to the ER by KDEL with somewhat lower efficiency due to the near-neutral (i.e. less acidic) pH in the proximal Golgi area.

In the ZnT5/6-silencing cells with normal $[Zn^{2+}]$ in the pre-*cis*-Golgi (Fig. 1d), ERp44 can likely bind Zn^{2+} and form complexes with clients in this compartment. However, since the labile $[Zn^{2+}]$ gets much lower in the *medial*-/*trans*-Golgi upon ZnT5/6-silencing (Fig. 1d), the ERp44-client complexes could partially be resolved when reaching this area, possibly allowing aberrant secretion of the clients. In this regard, we believe that Zn^{2+} in the distal-Golgi area serves as a gatekeeping factor for the ERp44-mediated client retention (please see also our response to comment 14 of Reviewer #1). On the other hand, the ZnT4-silencing elevated the labile $[Zn^{2+}]$ in the pre-*cis*-Golgi, promoting ERp44 to be relocated back to the ER immediately after reaching the *cis*-Golgi. In this situation, the Golgi-resident ERp44 molecules will be reduced in amount, again allowing greater client secretion than normal cells.

To facilitate readers' understanding, we have added these supplementary explanations to a Discussion section of the revised manuscript (lines 487–506).

8- A possible conclusion for ZnT4 is that its depletion accelerates the retrograde transport of ERp44. Looking at the data in Figure 6, this is the most likely explanation, and this is also what the authors propose. Is this a general effect, or is it specific for ERp44? This would also explain the higher Zn²⁺ levels in the ERGIC. The retrieval of ERp44 is dependent on KDEL-R and I find it hard to imagine that this effect is specific and has no effect on other clients

Response: To investigate if ZnT4-knockdown generally facilitates the Golgi-to-ER retrograde protein transport, we observed its effects on the intracellular localization of YFP-ERp44(3HA), an ERp44 mutant that lacks the Zn^{2+} -binding ability due to the mutations of the Zn^{2+} -binding histidine residues and hence displays no Zn^{2+} -dependent activation. Under normal conditions, YFP-ERp44(3HA) mostly accumulates in the Golgi,

as previously reported (Watanabe et al., 2019, *Nat. Commun.*). As demonstrated in new Supplementary Fig. 11, the knockdown of ZnT4 did little affect the predominant Golgi localization of YFP-ERp44(3HA), whereas YFP-ERp44(WT) displayed the reduced Golgi accumulation under this knockdown condition. These results suggest that the Zn²⁺-binding ability is necessary for ERp44 to show the ZnT4-dependent retrograde transport to the ER. Thus, the accelerated Golgi-to-ER transport by ZnT4-knockdown seems specific to ERp44 (WT). This experimental data has been additionally described in the revised manuscript (lines 241–249).

9- I think the conclusion the authors derive for Znt5/6 depletion in Figure 6 is unlikely to be correct. The authors conclude that ZnT5/6 depletion has no effect on ER export of ERp44. However, I think that a traffic defect out of the ER is the best explanation for the data. When using the ERp44 mutant lacking the RDEL motif (Fig 6D), there appears to be no GFP signal appearing in the Golgi area at any time point. The authors say that this is because the rate of transport to the Golgi is the same as out of the Golgi. I think this is not likely. More likely is a defect in trafficking out of the ER. I think it is required to stain for ERES markers and perform RUSH assays with other proteins in ZnT depleted cells to make the data less ambiguous.

Response: We again appreciate this critical comment. To examine if ZnT5/6 knockdown impairs the anterograde trafficking of ERp44, we analyzed the secretion of SBP-FLAG-ERp44(Δ RDEL) in the RUSH assay followed by western blotting for the culture medium, as shown in Supplementary Fig. 5C of the original manuscript. We repeated this experiment during the revision and reconfirmed that greater amount of SBP-FLAG-ERp44(Δ RDEL) was secreted out by the ZnT5/6 KD cells than by the siControl cells (new Supplementary Fig. 16C in the revised manuscript). Moreover, our live-cell imaging combined with the RUSH system revealed that ZnT5/6-silencing only marginally affected the rate of the ER-to-Golgi transport of SBP-Halo-ERp44(WT) (new Fig. 7e-i). Based on these results, it seems unlikely that ZnT5/6-silencing impaired the trafficking of ERp44 out of the ER. Instead, ZnT5/6-silencing seems to have greatly accelerated the Golgi-to-post Golgi trafficking of ERp44(Δ RDEL) without preventing its entry to the Golgi, leading to no apparent increase of ERp44(Δ RDEL) signals in the Golgi area. No defect

in the ER-to-Golgi transport under the ZnT5/6-silencing condition was also evident in observation with another model cargo protein, ST-SBP-EGFP (new Fig. 7a–d). This important issue has additionally been discussed in the Result section “Knockdowns of Golgi-resident ZnTs do not impair the ER-to-Golgi trafficking” of the revised manuscript (lines 394–428).

Overall, this is an important work, and these additional experiments would improve the overall quality and improve the clarity of the data and the conclusion derived from them.

Response: We greatly appreciate a number of constructive and useful comments provided by this reviewer. To address all the comments appropriately, we have performed a number of additional experiments and revised the text based on the results with greatest care. We do hope that this reviewer will find our revised manuscript much improved in terms of the clarity of the data and conclusions derived from them.

Reviewer #3:

This study provides an impressive set of data that reveals how ion transport, membrane trafficking and quality control processes cooperate to support important cellular function. Using a wide range of methods, the authors show a mechanistic link between the activities of Golgi-located Zn transporters ZnT4, ZnT5/ZnT6 and ZnT7 and regulation of ERp44-mediated proteostasis. The data provided suggest that: (i) ZnT4, ZnT5/ZnT6 and ZnT7 regulate sub-organelle Zn levels in trans-, medial and cis- (pre-cis-Golgi compartments); (ii) suppression of each ZnT has a different impact on the localization and activity of ERp44; and (iii) Zn²⁺ operates as a more dominant ERp44 regulator than pH. Altogether, the main findings of the paper uncover novel Zn-dependent mechanisms that regulate proteostasis in the early secretory pathway. Therefore, the manuscript might be of great interest to the readership of Nature Communications. However, in my view, the manuscript should be revised to address the following comments.

Response: We appreciate this positive overall evaluation.

1) This paper demonstrates that ZnT7 suppression inhibits Golgi-to-ER retrieval of ERp44 more than Znt5/6 or Znt4 knockdown. In light of this finding, why does ZnT7 silencing have a limited impact on ER retention of ERp44 substrates (ERAP1 or Ero1 α), while substrate secretion increases upon ZnT4 or ZnT5/6 depletion?

Response: We appreciate this essential comment. In ZnT7-silencing cells, ERp44 accumulates mostly in the Golgi apparatus, because the labile [Zn²⁺] is considerably reduced in the pre-*cis*-Golgi under this condition (Fig. 1d). Meanwhile, the [Zn²⁺] in the *cis*-/*medial*-/*trans*-Golgi is almost maintained after ZnT7-silencing (Fig. 1d), hence ERp44 can still function to retrieve client proteins from the Golgi to the ER. By contrast, ZnT4-silencing significantly reduced the amount of ERp44 localized in the Golgi (Fig. 3a), likely allowing more client proteins to proceed to the Golgi. ZnT5/6-silencing reduced the [Zn²⁺] in the *medial*-/*trans*-Golgi (Fig. 1d), not allowing the stable formation of ERp44-client complexes in this Golgi area. Under such circumstances, client proteins could be resolved from ERp44 and leaked out of the Golgi, resulting in the enhanced client secretion out of cells. Thus, we believe that ZnT4- and ZnT5/6-mediated Zn²⁺ in the *medial*-/*trans*-Golgi play essential roles for the ERp44-mediated client retention.

Based on the above findings, we carefully revised the corresponding parts of text so that the experimental data shown in Fig. 5 can be explained consistently with each other (lines 318–333). Please see also our responses to comment 14 of Reviewer #1 and comment 7 of Reviewer #2.

2) The authors claim that Zn prevails over pH as a determinant for Erp44 retrieval from Golgi to ER. In this context, it would be nice to evaluate how silencing of each ZnT affects the interaction between ERp44 and KDEL. I would like to encourage the authors to test this using immunoprecipitation experiments (or any other protein/protein interaction approach).

Response: Although we have so far paid much effort to detect the ERp44-KDEL interactions within cells and *in vitro*, we failed to detect the interaction between ERp44

and wild-type KDEL_R unfortunately. Therefore, we tested IP experiments using KDEL_R mutants known to constitutively retain in the Golgi under various conditions such as a various range of pH, KDEL-peptide activation, cross-linker treatment, zinc supplementation, and their combinations. However, none of the conditions enabled us to detect the interaction between ERp44 and KDEL_R. We believe that the complex formation between these two is so unstable and transient, and/or may require other mediating factor(s). We thus have to say reluctantly that we have given up obtaining direct evidence for the ERp44-KDEL_R interaction.

3) I have a concern regarding use of Giantin as a marker of medial Golgi for analysis of intra-Golgi distribution of halo-tagged constructs or ZnTs. EM data clearly suggest that Giantin is distributed across the rims of almost all Golgi cisternae, including cis and trans. It would be safer to use Man II where possible with ZnTs or halo-tag constructs. The authors may also consider using Golgin97 for trans-Golgi compartment labelling.

Response: As requested, we immunostained endogenous ManII and Golgin97 as a *cis*-/medial-Golgi and TGN markers, respectively, and used their signals to determine the localizations of the HaloTagged constructs with high accuracy. The new immunofluorescence data have been added to new Fig. 1b and 1c, and described in the revised text (lines 112–119).

4) The discussion of the cisternae maturation mechanism of ZnT distribution across the Golgi stack is confusing. Initially the authors say that ZnTs are excluded from the giantin-positive rims, where intra-Golgi trafficking of resident Golgi proteins occurs by Cop-I vesicles (according to the cisternae maturation model). Later in the discussion, they say that transporters are detectable in COP-I vesicles. These two statements contradict each other. I would avoid talking about cisterna maturation because the mechanisms of intra-Golgi trafficking still remain controversial. Instead, it would be better to discuss why some ZnTs prefer cis- or mid- or trans-Golgi. Is this defined by any specific signals, interactions or transmembrane domain length?

Response: We appreciate and accept this critical comment. We thus removed the discussions based on the cisternal maturation model. Instead, we have briefly discussed the possible involvement of the extra domains of ZnT4 and ZnT5 in determination of their intra-Golgi trafficking and specific partner proteins in the revised manuscript (lines 449–453).

5) Along the same line, I noted that in addition to the Golgi, ZnT4 also localizes at numerous peripheral spots (Fig. 3C). To which compartment do these spots correspond? This information might be important to understand if ZnT4 shows preferential affinity for trans-Golgi.

Response: As mentioned in our response to comment 7 of reviewer #1, we failed to detect the signals of endogenous ZnT4 by IF analysis, hence determined the localization of exogenously expressed ZnT4-FLAG. Co-immunostaining with several organelle markers revealed that majority of ZnT4-FLAG co-localized with early endosome, recycling endosome, and lysosome, whereas its minor portion was found at the Golgi apparatus and plasma membrane. This additional data is shown in new Supplementary Fig. 12, and described in the revised manuscript (lines 255–261).

6) Finally, it would probably be worth discussing the interplay between Golgi-associated ZnTs and ERp44-driven homeostasis in a broader physiological context. It is well known that loss of function of some ERp44 client proteins (like SUMF1) leads to very severe phenotypes. Some genetic disorders and/or mice phenotypes have been associated with ZnTs 4, 5 and 7 mutant and knockouts. Are there any features that are shared by ERp44 client- and ZnT-loss of function phenotypes? If not, could it be due to functional redundancy between Golgi-associated ZnTs?

Response: In response to this comment, we added discussion concerning the physiological relevance between the ERp44-KO and ZnT5-KO mice in the revised manuscript, as follows (lines 517–523).

ERp44-KO mice were reported to show growth retardation, cardiac developmental and

*functional defects and hypotension (Wang, et al., 2014, **J Am Heart Assoc**, Hisatsune et al., 2015, **Mol Cell**). Similarly, ZnT5-KO mice show growth retardation, and male-specific death because of the arrhythmias (Inoue et al., 2002, **Hum Mol Genet**). It is possible that ZnT5-KO inhibits function of ERp44 and allows aberrant secretion of male-specific ERp44-client proteins, leading to the heart failure. Secretome analyses with ZnT5-KO cells will identify proteins that may cause the above phenotypes, and further elucidate the physiological role of Zn²⁺-mediated protein quality control in the ESP.*

REVIEWERS' COMMENTS

Reviewer #1 (Remarks to the Author):

The authors have addressed all my comments. I just have a very minor comment: please correct the second sentence in the abstract to: Zn also regulates client binding...

Reviewer #2 (Remarks to the Author):

The authors have responded to all my previous criticism. I appreciate the hard work the authors have invested into revising the manuscript. I am happy with all revisions and have no further comments.

Reviewer #3 (Remarks to the Author):

The authors addressed all my concerns and suggestions and significantly improved the manuscript over the previous version. I suggest to accept the paper for publication.

Title: Zinc homeostasis governed by Golgi-resident ZnT family members regulates ERp44-mediated proteostasis at the ER-Golgi interface

Authors: Amagai Y. et al.

Manuscript No.: NCOMMS-21-48453

Reviewer #1 (Remarks to the Author):

The authors have addressed all my comments. I just have a very minor comment: please correct the second sentence in the abstract to: Zn also regulates client binding...

Response: We thank the reviewer for the positive evaluation of our work. As requested, we revised the second sentence in the original abstract by considering the context carefully, as follows: *ERp44, a chaperone operating in the early secretory pathway, also binds Zn²⁺ to regulate its client binding and release for the control of protein traffic and homeostasis.*

Reviewer #2 (Remarks to the Author):

The authors have responded to all my previous criticism. I appreciate the hard work the authors have invested into revising the manuscript. I am happy with all revisions and have no further comments.

Response: We thank the reviewer for the positive evaluation and appreciation of our work.

Reviewer #3 (Remarks to the Author):

The authors addressed all my concerns and suggestions and significantly improved the manuscript over the previous version. I suggest to accept the paper for publication.

Response: We thank the reviewer for the positive evaluation and recommendation of our work for publication in this journal.